# SpSC: A Fast and Provable Algorithm for Sampling-Based GNN Training

## Abstract

Neighbor sampling is a commonly used technique for training Graph Neural Networks (GNNs) on large graphs. Previous work has shown that sampling-based GNN training can be considered as Stochastic Compositional Optimization (SCO) problems and can be better solved by SCO algorithms. However, we find that SCO algorithms are impractical for training GNNs on large graphs because they need to store the moving averages of the aggregated features of all nodes in the graph. The moving averages can easily exceed the GPU memory limit and even the CPU memory limit. In this work, we propose a variant of SCO algorithms with sparse moving averages for GNN training. By storing the moving averages in the most recent iterations, our algorithm only requires a fixed size buffer, regardless of the graph size. We show that our algorithm preserves the convergence rate of the original SCO algorithm when the buffer size satisfies certain conditions. Our experiments validate our theoretical results and show that our algorithm outperforms the traditional Adam SGD for GNN training with a small memory overhead.

## 1 Introduction

Graph Neural Networks (GNNs) have become the state-of-the-art models for machine learning tasks on graph-structured data. By recursively aggregating the features of neighboring nodes, GNNs learn an embedding of the nodes and use the embedding for downstream tasks such as node classification (Kipf & Welling, 2017; Duran & Niepert, 2017) or link prediction (Zhang & Chen, 2017; 2018).

Due to the recursive neighbor aggregation, training GNNs on large graphs is computationally challenging. To alleviate the computation burden, various neighbor sampling methods have been proposed (Hamilton et al., 2017; Ying et al., 2018; Chen et al., 2018; Zou et al., 2019; Li et al., 2018; Chiang et al., 2019; Zeng et al., 2020). The idea is to compute an unbiased estimation of the aggregation result in each layer based on a sampled subset of neighbors. These sampling techniques enable GNN training on large graphs. However, due to the composition of the aggregation functions in multiple layers, the stochastic gradient obtained with sampled neighbor aggregation is not an unbiased estimation of the true gradient, which undermines the convergence property of SGD-based training algorithms.

Previous work has shown that sampling-based GNN training is actually a Stochastic Compositional Optimization (SCO) problem (Cong et al., 2020; 2021). Cong et al. (2021) show that SCO algorithms can achieve faster convergence than the commonly used Adam SGD for GNN training on small graphs. Despite their good convergence property, SCO algorithms are not widely adopted for GNN training due to two reasons. First, although SCO algorithms achieve smaller training losses, the obtained GNN models usually have poor generalization – the validation and test accuracy are lower than the models trained by Adam SGD. Second, SCO algorithms need to maintain the moving averages of aggregation results of all nodes in the graph. For large graphs, the moving averages may exceed the memory capacity of the GPU. While it is possible to store the moving averages in CPU memory, copying the data from CPU to GPU in each iteration is expensive, which may negate the benefits of the faster convergence of SCO algorithms.

To address the above issues, we propose a Sparse Stochastic Compositional (SpSC) gradient method in this work. Our main idea is to store the moving averages for nodes sampled in the most recent

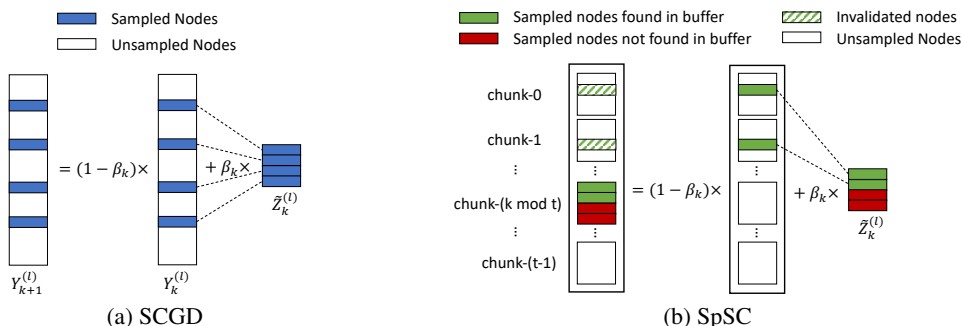

(a) SCGD                                                (b) SpSC

Figure 1: Updating the moving average of $\widetilde{Z}^{(l)}$. SCGD needs to store the moving averages for all nodes in the graph. In SpSC, we only stores the data for nodes sampled in the past $t$ iterations.

iterations instead of all nodes. As only a small number of nodes are stored, our algorithm has small memory consumption even for large graphs. We provide a convergence analysis on SpSC and show that, when the number of stored iterations satisfies certain constraints, SpSC can preserve the asymptotic convergence rate of the original SCO algorithm. In practice, the sparse moving averaging slightly slows down the convergence of SCO algorithm, but it surprisingly overcomes its poor generalization problem and achieves higher accuracy for sampling-based GNN training. Compared with Adam SGD, our algorithm incurs a small overhead for updating the moving averages in each iteration, but the overhead can be easily justified by the faster convergence of our algorithm.

Our experiments with two GNN models on different input graphs validate our theoretical results and show that our algorithm achieves higher accuracy than Adam SGD with the same or less amount of training time.

## 2 BACKGROUND AND MOTIVATION

To facilitate our discussion, we first give background on sampling-based GNN training and its relation to stochastic compositional optimization.

### 2.1 GNN COMPUTATION

The computation at each layer of a GNN is conducted in two steps: `aggregate` and `update`. For each node $v$ in the graph, the `aggregate` function gathers data from its neighboring nodes and returns the aggregation result as

$$z_v = \text{Agg}_v(h_{ne[v]}, x_v, x_{ne[v]}). \tag{1}$$

Here, $h_{ne[v]}$ is the intermediate features of $v$'s neighbors from the previous layer, $x_v$ is the input feature of $v$, and $x_{ne[v]}$ is the input features of $v$'s neighbors. The `update` function uses the aggregated value to produce the intermediate features of $v$ as

$$h_v = \text{Upd}_v(z_v, x_v). \tag{2}$$

By stacking the intermediate features and the input features of all nodes, the computation at layer $l$ can be written as

$$Z^{(l)} = \text{Agg}(H^{(l-1)}, X), \qquad H^{(l)} = \text{Upd}(Z^{(l)}, X, W^{(l)}). \tag{3}$$

Here, $H^{(l-1)} = [h_1^{(l-1)}, \ldots, h_N^{(l)}]$ denotes the intermediate features of all nodes at layer $l-1$, $Z^{(l)} \in \mathbb{R}^{N \times d_l}$ is the aggregated features of all nodes at layer $l$, $X = [x_1, \ldots, x_N]$ is the input features of all nodes, and $W^{(l)}$ is the learnable weights. As an example, Graph Convolutional Network (GCN) (Kipf & Welling, 2017) has $\text{Agg}(H^{(l-1)}, X) = PH^{(l-1)}$ where $P$ is the normalized Laplacian matrix of the graph, and $\text{Upd}(Z^{(l)}, X, W^{(l)}) = \sigma(Z^{(l)}W^{(l)})$ where $\sigma$ is a non-linear activation function. Many other GNNs can be expressed in this form with different definitions of Agg and Upd (Zhou et al., 2018).

## 2.2 Sampling-based GNN Training as Stochastic Compositional Optimization

When the graph is large, the neigbhbor aggregation operation $\mathrm{Agg}$ incurs a large overhead, making the training of GNNs computationally challenging. Therefore, prior work has proposed to replace the $\mathrm{Agg}$ function with a sampled neighbor aggregation operation $\widetilde{\mathrm{Agg}}$. By sampling the neighboring nodes, an unbiased estimate of $Z^{(l)}$ is computed at each layer, i.e.,

$$\widetilde{Z}^{(l)} = \widetilde{\mathrm{Agg}}(H^{(l-1)}, X) \tag{4}$$

with $\mathbb{E}[\widetilde{Z}^{(l)}] = Z^{(l)}$. If we define the computation at layer $l$ of the original GNN as a function

$$f^{(l)}(Z^{(l-1)}, W^{(l-1)}, ..., W^{(T)}) = [Z^{(l)}, W^{(l)}, ..., W^{(T)}] \tag{5}$$
$$= [\mathrm{Agg}(\mathrm{Upd}(Z^{(l-1)}, X, W^{(l-1)})), W^{(l)}, ..., W^{(T)}]],$$

the computation with sampled neighbor aggregation can be written as a stochastic function

$$f_{\xi_l}^{(l)}(\widetilde{Z}^{(l-1)}, W^{(l-1)}, ..., W^{(T)}) = [\widetilde{Z}^{(l)}, W^{(l)}, ..., W^{(T)}] \tag{6}$$
$$= [\widetilde{\mathrm{Agg}}(\mathrm{Upd}(\widetilde{Z}^{(l-1)}, X, W^{(l-1)})), W^{(l)}, ..., W^{(T)}]]$$

where $\xi_l$ represents the sampled neighbors at layer $l$. Since $\widetilde{Z}^{(l)}$ is an unbiased estimate of $Z^{(l)}$, we have $\mathbb{E}[f_{\xi_l}^{(l)}] = f^{(l)}$, and the computation of a $T$-layer GNN can be written as

$$F(\theta) = \mathbb{E}_{\xi_{T+1}} \left[ f_{\xi_{T+1}}^{(T+1)} \left( \mathbb{E}_{\xi_T} \left[ f_{\xi_T}^{(T)} \left( ...E_{\xi_1}[f^{(1)}(\theta)]... \right) \right] \right) \right] \tag{7}$$

where $\theta = [X, W^{(1)}, ..., W^{(T)}]$, $f^{(T+1)}$ is the loss function, and $f_{\xi_{T+1}}^{(T+1)}$ corresponds to the estimated loss with mini-batch sampling. Note that we put all the learnable weights in $\theta$ to formulate the computation as a stochastic compositional function. Our goal is to minimize $F(\theta)$, which is exactly a multi-level SCO problem.

## 2.3 Large Memory Consumption Issue with A Naive Implementation

SCO has been well studied in the past few years, and many algorithms with guaranteed convergence have been proposed (Zhang & Xiao, 2019; Yang et al., 2019; Chen et al., 2020; Yang et al., 2019; Balasubramanian et al., 2020; Chen et al., 2020; Lian et al., 2017; Wang et al., 2017b; Ghadimi et al., 2020). It seems straightforward to adopt these SCO algorithms to achieve faster training of GNNs. However, these algorithms have large memory consumption when applied to GNN training and cannot run on GPUs for large graphs.

To see the problem, let us consider the implementation of the SCGD algorithm (Yang et al., 2019) for GNN training. Formally, the algorithm is written as

$$y_{k+1}^{(1)} = (1 - \beta_k) y_k^{(1)} + \beta_k f_{\xi_{1,k}}^{(1)}(\theta_k), \tag{8}$$

$$y_{k+1}^{(l)} = (1 - \beta_k) y_k^{(l)} + \beta_k f_{\xi_{l,k}}^{(l)}(y_{k+1}^{(l-1)}), \quad 2 \leq l \leq T, \tag{9}$$

$$\theta_{k+1} = \theta_k - \alpha_k \nabla f_{\xi_{1,k}}^{(1)}(\theta_k) \nabla f_{\xi_{2,k}}^{(2)}(y_k^{(1)}) \ldots \nabla f_{\xi_{T+1,k}}^{(T+1)}(y_k^{(T)}). \tag{10}$$

The key idea is to store an auxiliary variable $y^{(i)}$ to maintain the moving average of each composite function. Since $f_{\xi_l}^{(l)}$ returns the exact values of $W^{(l)}, ..., W^{(T)}$, we only need to maintain a moving average of $\widetilde{Z}^{(l)}$ for each layer. The computation is shown in Figure 1a. The moving average of the aggregated features is stored in $Y^{(l)}$ with each row for one node. In each iteration, some nodes (rows) are sampled, and the estimated aggregation results $\widetilde{Z}^{(l)}$ are merged into $Y^{(l)}$ based on Formula (8) and (9). For the nodes that are not sampled, we simply multiply the corresponding rows of $\bar{Z}^{(l)}$ by $(1 - \beta_k)$. Since the number of rows in $Y^{(l)}$ is the number of nodes in the graph, $Y^{(l)}$ takes a lot of memory when the graph is large. For example, for training a 3-layer GCN on a graph with two million nodes, suppose the hidden state dimension $d_l = 512$ and a floating point has 4 bytes, $Y$ takes $3 \times 2\mathrm{M} \times 512 \times 4 = 12\mathrm{GB}$ of memory. All of the existing SCO algorithms need to maintain this moving average, which impedes their application to large-scale GNN training.

## 3   SPARSE STOCHASTIC COMPOSITIONAL GRADIENT DESCENT

To reduce the memory consumption of SCO algorithms for GNN training, we propose a Sparse Stochastic Compositional (SpSC) gradient method. Instead of storing the moving averages of all nodes in the graph, we only store the moving averages of nodes that are sampled in the most recent iterations.

As shown in Figure 1b, we maintain a fixed size buffer of the moving averages. The buffer is divided into $t$ chunks with each chunk for the $\widetilde{Z}^{(l)}$ of one iteration. The size of each chunk is $m_l \cdot d_l$ where $m_l$ is the maximum number of the nodes that can be sampled at layer $l$ and $d_l$ is the hidden state dimension. Initially, the buffer is empty. In every iteration, we first check if the sampled nodes are in the buffer. For the nodes that are found in the buffer, we collect the corresponding rows of the buffer and add them to $\widetilde{Z}_k^{(l)}$ based on Formula (8) and (9). For the nodes that are not found in the buffer, we multiply the corresponding rows of $\widetilde{Z}_k^{(l)}$ by $\beta_k$. The updated $\widetilde{Z}_k^{(l)}$ is then written to chunk-$(k \bmod t)$. All the other chunks are multiplied by $(1 - \beta_k)$. Since the sampled nodes found in the buffer are updated to chunk-$(k \bmod t)$, the original values in chunk-0 and chunk-1 are invalidated, as shown by the shadowed rows in Figure 1b. As the buffer size is a constant ($T \cdot t \cdot m_l \cdot d_l$) regardless of the graph size, our algorithm can be employed to train GNN on very large graphs.

Our algorithm overwrites chunk-$(k \bmod t)$ in iteration $k$. The information of the overwritten nodes is lost. The update of the moving averages can be written as

$$y_{k+1}^{(l)} = (1 - \beta_k)y_k^{(l)} + \beta_k f_{\xi_{l,k}}^{(l)}(y_{k+1}^{(l-1)}) - \prod_{j=k-t+1}^{k-1}(1 - \beta_j)u_k^{(l)} \tag{11}$$

where

$$u_k^{(l)} = P(\xi_{l,k-t}/(\xi_{l,k-t+1} \cup \cdots \cup \xi_{l,k}))y_{k-t+1}^{(l)}. \tag{12}$$

$P(\xi_{l,k-t}/(\xi_{l,k-t+1} \cup \cdots \cup \xi_{l,k}))$ is a matrix with binary values representing the overwritten nodes in the current iteration $k$, i.e., the nodes that are sampled in iteration $k - t$ and are not sampled in the following $t$ iterations. Since we only store the moving averages of nodes that are sampled in the most recent $t$ iterations, the information of the overwritten nodes is lost. These nodes are multiplied by $(1 - \beta_j)$ in every iteration after iteration $k - t$. The values of the overwritten rows are $\prod_{j=k-t+1}^{k-1}(1 - \beta_j)u_k^{(l)}$. Our algorithm simply replaces Formula (8) and (9) in the SCGD algorithm with Formula (11).

To study the convergence property of SpSC, we make the following assumptions that are commonly used in the analysis of SCO algorithms (Yang et al., 2019; Balasubramanian et al., 2020).

**Assumption 1.** *The composite functions $f^{(l)}$ are $L_l$-smooth. That is, for any $y$ and $y'$, we have $\|\nabla f_{\xi_l}^{(l)}(y) - \nabla f_{\xi_l}^{(l)}(y')\| \leq L_l\|y - y'\|$.*

**Assumption 2.** *The stochastic gradients of the composite functions $f^{(l)}$ are bounded in expectation, i.e., $\mathbb{E}[\|\nabla f_{\xi_l}^{(l)}(y)\|^2] \leq C_l^2$.*

**Assumption 3.** *The estimated aggregation results obtained by sampled neighbor aggregation is unbiased, i.e., $\mathbb{E}[f_{\xi_l}^{(l)}(y)] = f^{(l)}(y)$, and the stochastic gradient of $f^{(l)}$ is unbiased, i.e., $\mathbb{E}[\nabla f_{\xi_l}^{(l)}(y)] = \nabla f^{(l)}(y)$.*

Following the single-timescale analysis of the algorithm (Balasubramanian et al., 2020), we use large batches for estimating the composite functions and assume that the estimation variance is small.

**Assumption 4.** *The estimated aggregation results have small bounded variance, i.e., $\mathbb{E}[\|f_{\xi_l,k}^{(l)}(y) - f^{(l)}(y)\|] \leq \beta_k V^2$.*

This is a reasonable assumption for GNN training on GPUs as we always sample a batch of nodes for neighbor aggregation to achieve better utilization of the GPU parallelism.

In additional to the conventional assumptions, we make an assumption on the moving averages.

**Assumption 5.** *The moving average of the aggregated features are bounded, i.e., $\mathbb{E}[\|y^{(l)}\|^2] \leq D^2$.*

The convergence rate of our algorithm is summarized in the following theorem.

**Theorem 1.** *Under Assumptions 1-5, if we set $\beta \leq 1 - 2^{(-1/(2T-1))}$ and $\alpha = \beta \cdot \min\left(\frac{1}{2C_1 T}\sqrt{\frac{2(1-\beta)^{2T-1}-1}{C_{max}}}, \frac{2}{\sum_{l=1}^{T} A_l^2 + 16C_{max}C_1^2 T}\right)$, the model parameters $\{\theta_k\}$ of our training algorithm with (11) for updating sparse moving averages satisfy*

$$\frac{1}{K}\sum_{k=0}^{K-1} \mathbb{E}\left[\|\nabla F(\theta_k)\|\right]^2 \leq O(\beta) + 4C_{max}T^2\left(\frac{(1+C_{max})(1-\beta)^{2(t-1)}}{\alpha\beta^2} + \frac{(1-\beta)^{2t+1}}{T\alpha\beta^2}\right)D^2 \tag{13}$$

*where $C_{max} = max(C_2^2 C_3^2 \cdots C_l^2, C_3^2 C_4^2 \cdots C_l^2, \cdots, C_l^2, 1)$ for $l = 2\ldots T$, $t$ is the number of buffer chunks used in our algorithm.*

The last term on the RHS of (13) reveals how the convergence of the SCGD algorithm is affected by the sparse moving averages. The larger $t$ we use (i.e., the more chunks we have in the buffer), the smaller $(1-\beta)^{2(t-1)}$ and $(1-\beta)^{2t+1}$ we have, and the faster convergence we achieve. In theory, if we can make $(1-\beta)^{2(t-1)} = O(\beta^4)$, the algorithm will achieve $O(\sqrt{1/K})$ convergence rate. As $\beta \to 0$, we need larger $t$ to maintain the convergence rate, and eventually, we will need to store the moving average of all nodes in the graph.

**Applying Sparse Moving Averages to SCSC.** Our sparse moving average can also be applied to other SCO algorithms. For example, the Stochastically Corrected Stochastic Compositional gradient method (SCSC) (Chen et al., 2020) has a correction term in the update of the moving averages. Because of the correction term, SCSC needs a relaxed assumption on the estimation error of the composite functions. More specifically, if we change (11) to

$$y_{k+1}^{(l)} = (1 - \beta_k)y_k^{(l)} + f_{\xi_{l,k}}^{(l)}(y_{k+1}^{(l-1)}) - (1-\beta_k)f_{\xi_{l,k}}^{(l)}(y_k^{(l-1)}) - \prod_{j=k-t+1}^{k-1}(1-\beta_j)u_k^{(l)} \tag{14}$$

and replace (8) and (9) with (14), we can relax Assumption 4 as

**Assumption 6.** *The estimated aggregation results have bounded variance, i.e., $\mathbb{E}[\|f_{\xi_{l,k}}^{(l)}(y) - f^{(l)}(y)\|] \leq V^2$.*

The convergence rate of SCSC with sparse moving averages is summarized as follows.

**Theorem 2.** *Under Assumptions 1-3 and 5-6, if we choose $\alpha_k = \alpha = \frac{c_\alpha}{\sqrt{K}}$ and $\beta_k = \beta = \frac{c_\beta}{\sqrt{K}}$, the model parameters $\{\theta_k\}$ of SCSC with (14) for updating sparse moving averages satisfy*

$$\frac{1}{K}\sum_{k=0}^{K-1}\mathbb{E}[\|\nabla F(\theta_k)\|^2] \leq O(\beta) + \frac{2(1+\beta)(1-\beta)^{2(t-1)}}{\alpha\beta}\sum_{l=1}^{T}D_l^2$$
$$+ \frac{6(1-\beta)^{2(t-1)}}{\alpha}C_u\sum_{l=1}^{T}D_l^2 \tag{15}$$

*where $C_u = max\left(4C_l^2 + \gamma_l\right)$ for $l = 1\ldots T$.*

The result suggests that, if we can set $t$ such that $(1-\beta)^{2(t-1)} = O(\beta^3)$, the algorithm will achieve $O(\sqrt{1/K})$ convergence rate.

## 4 IMPLEMENTATION DETAILS

Algorithm 1 describes an implementation of Formula (11) in our algorithm. For each layer, we allocate a buffer ($buf$) of size $tm_l \times d_l$ for the moving averages. The buffered nodes are maintained in a list $node\_list \in \mathbb{R}^{tm_l}$ where $node\_list[i]$ is the index of the node stored at $buf[i]$. If $buf[i]$ is empty, $node\_list[i]$ is set to -1. In every iteration $k$, we first get the location of chunk-($k \mod t$). Then, we look up each of the sampled nodes in the buffer (line 3). The LookUp function computes

---

**Algorithm 1:** Updating sparse moving average of aggregated features at layer $l$ in iteration $k$

---

**Input:** Sampled nodes $S$, Buffered nodes $node\_list \in \mathbb{R}^{tm_l}$, $buf^{(l)} \in \mathbb{R}^{tm_l \times d_l}$, $\widetilde{Z}_k^{(l)} \in \mathbb{R}^{|S| \times d_l}$

    `// Get the location of chunk-(`$k \bmod t$`)`

1  $start = (k \bmod t) * m_l$;

2  $end = chunk\_start + |S|$;

    `// Look up the sampled nodes in the buffer`

3  $idx\_in\_buf, idx\_in\_z = \mathrm{LookUp}(S, node\_list)$;

    `// Update the moving average for the sampled nodes`

4  $\widetilde{Z}_k^{(l)} = \beta_k * \widetilde{Z}_k^{(l)}$;  $\widetilde{Z}_k^{(l)}[idx\_in\_z] = (1 - \beta_k) * buf^{(l)}[idx\_in\_buf] + \widetilde{Z}_k^{(l)}[idx\_in\_z]$;

    `// Update the moving average for all buffered nodes`

5  $buf^{(l)} = (1 - \beta_k) * buf^{(l)}$;  $buf^{(l)}[start : end] = \widetilde{Z}_k^{(l)}$;

    `// Invalidate the old buffer for the sampled nodes`

6  $node\_list[idx\_in\_buf] = -1$;

    `// Add the sampled nodes to` *node_list*

7  $node\_list[start : end] = S$;

---

the intersection of $S$ and $node\_list$ and returns the indices of the overlapping nodes in the two arrays. If a sampled node is not found in the buffer, we multiply the corresponding row of $\widetilde{Z}_k^{(l)}$ by $\beta_k$. If a sampled node is in the buffer, we read in its current moving average and update the corresponding row of $\widetilde{Z}_k^{(l)}$ (line 4). For buffered nodes that are not sampled, we simply multiply their moving averages by $(1 - \beta_k)$ (line 5). For buffered nodes that are sampled, we invalidate their original buffer by setting $node\_list[idx\_in\_buf]$ to $-1$ (line 6). Last, we add the sampled nodes to the $node\_list$.

Most of the operations in Algorithm 1 are simple vector operations, and they incur little overhead. The performance bottleneck is the $\mathrm{LookUp}$ function. With a naive implementation, it has $O(tm_l \log |S|)$ time complexity, assuming $S$ is sorted. In our implementation, we use an auxiliary array $node\_loc \in \mathbb{R}^N$ to store the locations of all nodes in the buffer and accelerate the $\mathrm{LookUp}$ function. Specifically, if node-$i$ is in the buffer, we store its location in buffer in $node\_loc[i]$; otherwise, $node\_loc[i]$ is set to -1. With the auxiliary array, the $idx\_in\_z$ can be obtained by comparing $node\_loc[S]$ with zero, and the $idx\_in\_buf$ is simply $node\_loc[S][idx\_in\_z]$. Before updating the $node\_list$ at line 7 of Algorithm 1, we remove the overwritten nodes from $node\_loc$ by setting $node\_loc[node\_list[start : end]]$ to -1. Finally, we store the locations of the newly sampled nodes to $node\_loc$ by setting $node\_loc[S]$ to $[start, start + 1, \ldots, end]$. It is easy to see that all these operations have $O(|S|)$ time complexity.

## 5 EVALUATION

### 5.1 EXPERIMENTAL SETUP

We conduct our experiments on a workstation with an Nvidia RTX 3090 GPU, an Intel Xeon Gold 6226R CPU, and 512GB RAM. Our code is implemented with PyTorch 1.8.0 and PyTorch Geometric 1.7.0.

We evaluate our algorithm on five graphs as listed in Table 1. The `reddit` and `yelp` graph are adopted from GraphSAINT (Zeng et al., 2020), and the `arxiv`, `proteins`, `products` are from the Open Graph Benchmark (Hu et al., 2020).

We apply our algorithm to two GNN models: GCN (Kipf & Welling, 2017) and GraphSAGE (Hamilton et al., 2017). Both models have three convolutional layers. We use Formula (11) for updating the moving average instead of Formula (14). This is because Formula (14) requires two forward passes which incurs extra overheads. The algorithm is run for 50 epochs. We set $\beta$ to 0.2 initially, and decrease it to 0.1 at epoch 20, and further decrease it to 0.05 at epoch 40. The number of buffered chunks ($t$) is set to 8. We adopt the layer-wise sampling method in Zou et al. (2019) for neighbor sampling. The batch size is set to 4096, and the number of sampled neighbors in each layer is set to 8192.

### 5.2 TRAINING RESULTS

**Validation Accuracy.** Figure 2a shows the validation accuracy of different algorithms for training a GCN on `arxiv`. We can see that full neighbor aggregation (Adam_Full) achieves the highest

Table 1: Graph datasets ('m' stands for multi-label classification).

|          | reddit | yelp    | arxiv | proteins | products |
|----------|--------|---------|-------|----------|----------|
| #nodes   | 233K   | 717K    | 169K  | 132K     | 2.4M     |
| #edges   | 11.6M  | 7.0M    | 1.2M  | 79M      | 123M     |
| #classes | 41     | 100 (m) | 40    | 112 (m)  | 47       |

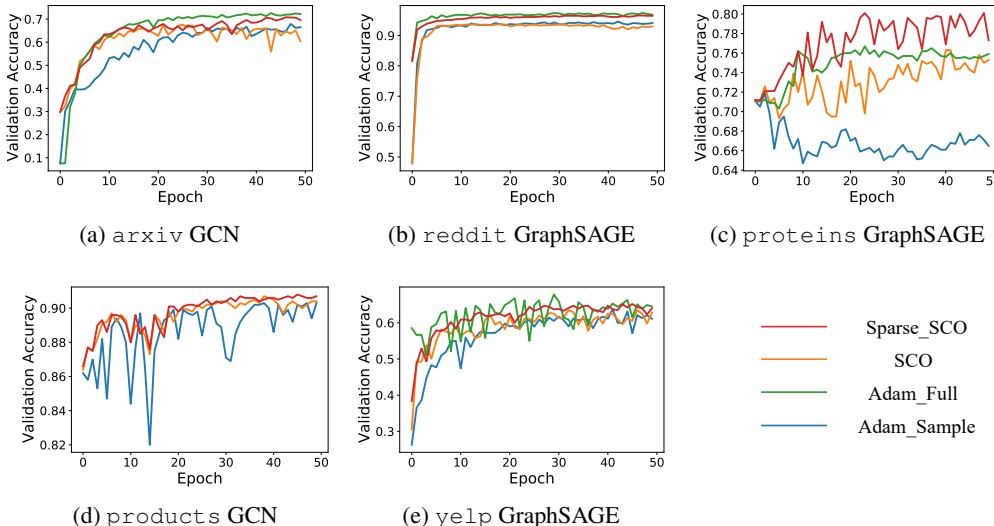

(a) `arxiv` GCN      (b) `reddit` GraphSAGE      (c) `proteins` GraphSAGE

(d) `products` GCN      (e) `yelp` GraphSAGE

Figure 2: Validation accuracy over epochs.

accuracy. This is reasonable because full neighbor aggregation returns unbiased estimates of gradients. If neighbor sampling is used, the accuracy clearly drops with the same training algorithm (Adam_Sample). Our algorithm (Sparse_SCO) is able to improve the convergence of sampling-based GNN training and achieves almost the same accuracy as full neighbor aggregation. The results on reddit and yelp are similar. On products graph, Adam_Full runs out of memory, so we only show the results of sampling-based training in Figure 2d. Interestingly, we find that our algorithm with neighbor sampling achieves even higher accuracy than Adam_Full on proteins graph, as shown in Figure 2c. The original SCO algorithm (which stores the moving averages for all nodes in the graph) also has lower accuracy than Sparse_SCO. This is probably due to the overfitting of models by Adam_Full and SCO.

**Training Loss.** Figure 3 shows the training loss of different algorithms on different graphs. For reddit and yelp, we are able to run full neighbor aggregation on our GPU. As expected, Adam_Full achieves the smallest training loss. Adam_Sample, however, has the slowest convergence. SCO achieves training loss close to Adam_Full. We run our algorithm with different $t$'s. The larger $t$ we use, the smaller training loss we obtain. The results are consistent with our theoretical analysis and also suggesting that the poor accuracy of the original SCO algorithm is probably due to overfitting. For products graph, we are not able to run full neighbor aggregation. The results show a clearly faster convergence of our algorithm than Adam SGD for sampling-based training.

**Test Accuracy.** Table 2 lists the test accuracy of the models trained by different algorithms. For reddit and yelp, we follow the GraphSAINT paper (Zeng et al., 2020) and report the F1-micro score. For proteins, we follow the OGB (Hu et al., 2020) and report the ROC-AUC. We can see that our training algorithm achieves the highest test accuracy for both GCN and GraphSAGE on almost all the graphs. We do not include the results for GCN on yelp graph because its accuracy is apparently lower than GraphSAGE with all training algorithms, probably due to the limited expressiveness of the GCN model. While it is hard to draw a direct comparison with GraphSAINT because the sampling methods and the model architectures are different, our test accuracy on reddit and yelp matches the best accuracy reported by GraphSAINT (Zeng et al., 2020). It is worth noting that our algorithm achieves higher accuracy than Adam SGD with both full neighbor aggregation and neighbor sampling on proteins and products graph. The numbers are higher than the accuracy

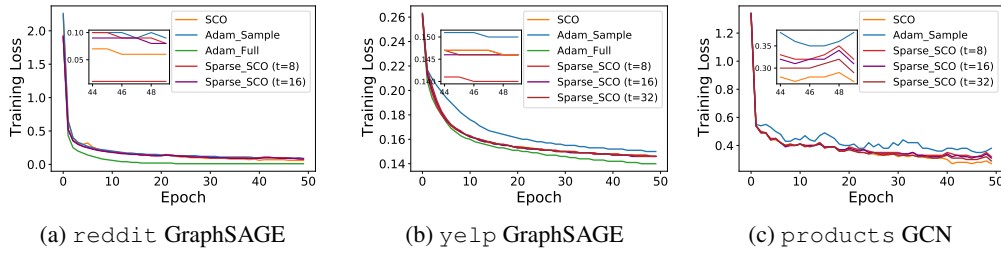

Figure 3: Training loss over epochs.

Table 2: Test accuracy of models trained by different algorithms on different graphs ('-' means not available due to out of memory).

(a) GCN

|            | reddit | arxiv | proteins | products |
|------------|--------|-------|----------|----------|
| Adam_Full  | 0.961  | **0.712** | 0.749    | -        |
| Adam_Sample| 0.957  | 0.663 | 0.726    | 0.790    |
| SCO        | 0.945  | 0.698 | 0.744    | 0.773    |
| Sparse_SCO | **0.961** | 0.711 | **0.770** | **0.802** |

(b) GraphSAGE

|            | reddit | yelp  | arxiv | proteins | products |
|------------|--------|-------|-------|----------|----------|
| Adam_Full  | 0.963  | 0.632 | **0.714** | 0.758    | -        |
| Adam_Sample| 0.956  | 0.631 | 0.685 | 0.718    | 0.787    |
| SCO        | 0.913  | 0.628 | 0.644 | 0.717    | -        |
| Sparse_SCO | **0.966** | **0.651** | 0.713 | **0.779** | **0.801** |

of the same models reported in OGB Leaderboards (ogb, 2021). The results suggest that there is an improvement space for the accuracy of GNN models by using better training algorithms.

**Execution Time.** Table 3 lists the time per epoch of different training algorithms on different graphs. Although full neighbor aggregation achieves good convergence in many cases, it incurs a much large computation overhead than sampled neighbor aggregation. The execution time of Adam_Full is 7x to 55x longer than that of Adam_Sample. Compared with Adam_Sample, the SCO algorithm incurs a small overhead for updating the moving average of aggregation results. Our algorithm runs slightly slower than SCO because the LookUp operation in Algorithm 1 incurs an extra overhead. However, the execution time is still much smaller than full neighbor aggregation.

**Memory Consumption.** Figure 4 shows the memory consumption of different algorithms. We collect the numbers by calling the max_memory_allocated function in PyTorch at the end of the first epoch. For arxiv, reddit, proteins, and yelp, full neighbor aggregation takes much larger memory space than sampled aggregation. SCO and Sparse_SCO require additional memory for storing the moving averages. Because we only store the moving average of nodes sampled in recent iterations, Sparse_SCO uses less memory than SCO. On products graph, SCO requires extremely large memory due to the massive number of nodes, while our Sparse_SCO can run with only 2GB of GPU memory.

## 6  RELATED WORK

To overcome the scalability limitation of GNN training, various neighbor sampling methods have been proposed, including node-wise sampling (Hamilton et al., 2017; Ying et al., 2018), layer-wise sampling (Chen et al., 2018; Huang et al., 2018; Zou et al., 2019), and subgraph sampling (Zeng et al., 2020; Chiang et al., 2019). These sampling-based training methods achieve good accuracy in practice (particularly on small graphs), but they lack theoretical justification.

Table 3: Time per epoch of different training algorithms in seconds ('-' means not available due to out of memory).

(a) GCN

|  | reddit | arxiv | proteins | products |
|---|---|---|---|---|
| Adam_Full | 28.4 | 1.97 | 20.65 | - |
| Adam_Sample | 0.53 | 0.28 | 0.55 | 1.01 |
| SCO | 0.72 | 0.37 | 0.67 | 2.31 |
| Sparse_SCO | 0.82 | 0.51 | 0.81 | 1.24 |

(b) GraphSAGE

|  | reddit | yelp | arxiv | proteins | products |
|---|---|---|---|---|---|
| Adam_Full | 45.29 | 23.07 | 4.10 | 46.16 | - |
| Adam_Sample | 0.82 | 2.38 | 0.50 | 0.93 | 1.47 |
| SCO | 1.09 | 3.61 | 0.61 | 1.03 | - |
| Sparse_SCO | 1.26 | 2.93 | 0.85 | 1.26 | 1.87 |

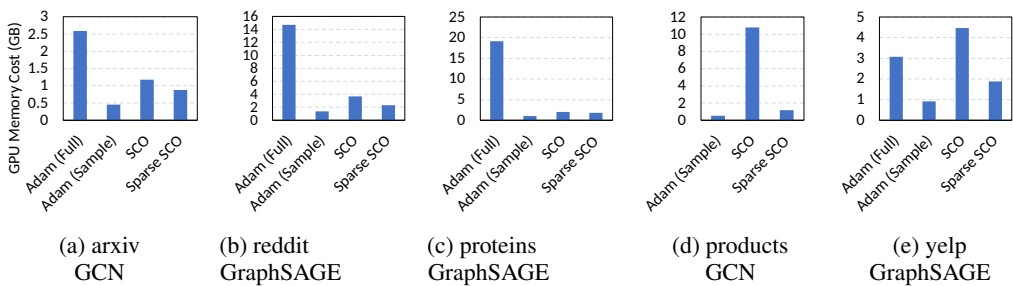

| (a) arxiv GCN | (b) reddit GraphSAGE | (c) proteins GraphSAGE | (d) products GCN | (e) yelp GraphSAGE |
|---|---|---|---|---|

Figure 4: GPU memory consumption of different algorithms.

Some recent works (Cong et al., 2020; 2021) point out that sampling-based GNN training is actually multi-level Stochastic Compositional Optimization (SCO). However, they either use this connection to justify their sampling techniques and fall back to Adam SGD for training (Cong et al., 2020), or they directly adopt an SCO algorithm without considering the large memory consumption issue (Cong et al., 2021).

The research on SCO traces back to (Ermoliev, 1976) where a two-timescale stochastic approximation scheme was proposed for two-level problems. Various SCO algorithms and convergence analyses have been proposed ever since (Wang et al., 2017a;b; Lian et al., 2017; Yang et al., 2019; Zhang & Xiao, 2019; Chen et al., 2020; Balasubramanian et al., 2020; Ghadimi et al., 2020; Hu et al., 2019; Lin et al., 2018). Despite a substantial volume of work on SCO in recent years, none of the existing work has considered the large memory consumption issue and the data movement overhead of the algorithms. Our work is the first to establish a convergence analysis for SCO algorithms with sparse moving averages.

## 7 CONCLUSION

In this work, we propose a new variant of SCO algorithm for training graph neural networks on large graphs. Our main idea is to maintain a sparse moving average of the aggregation results in each convolutional layer. We study the convergence property of our algorithm and show that the algorithm can achieve $O(\sqrt{1/K})$ convergence rate when a sufficient amount of moving averages are maintained. Our experiments with two GNN models on different graphs validate our theoretical results and show a clear advantage of our algorithm against Adam SGD for GNN training.

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

# 8 APPENDIX

Link to our source code: `https://anonymous.4open.science/r/iclr2022_artifact-0FE6/`

## 8.1 PROOF TO THEOREM 1

Based on the discussion in Section 3, our proposed algorithm can be written as

$$y_{k+1}^{(1)} = (1 - \beta_k)y_k^{(1)} + \beta_k f_{\xi_{1,k}}^{(1)}(\theta_k) - \prod_{j=k-t+1}^{k-1}(1 - \beta_j)u_k^{(1)}, \tag{16}$$

$$y_{k+1}^{(l)} = (1 - \beta_k)y_k^{(l)} + \beta_k f_{\xi_{l,k}}^{(l)}(y_{k+1}^{(l-1)}) - \prod_{j=k-t+1}^{k-1}(1 - \beta_j)u_k^{(l)}, \quad 2 \le l \le T, \tag{17}$$

$$\theta_{k+1} = \theta_k - \alpha_k \nabla_k, \tag{18}$$

with

$$u_k^{(l)} = P(\xi_{l,k-t}/(\xi_{l,k-t+1} \cup \cdots \cup \xi_{l,k}))y_{k+t-1}^{(l)} \tag{19}$$

and

$$\nabla_k = \nabla f_{\xi_{1,k}}^{(1)}(\theta_k) \cdots \nabla f_{\xi_{T,k}}^{(T)}(y_{k+1}^{(T-1)}) \nabla f_{\xi_{T+1,k}}^{(T+1)}(y_{k+1}^{(T)}). \tag{20}$$

### 8.1.1 SUPPORTING LEMMA

**Lemma 3.** *The change of $y_k^{(l)}$ in every iteration is bounded, i.e.,*

$$\mathbb{E}\left[\left\|y_{k+1}^{(1)} - y_k^{(1)}\right\|^2\right] \le \frac{\beta_k^2}{1 - \beta_k}\mathbb{E}\left[\left\|f^{(1)}(\theta_k) - y_{k+1}^{(1)}\right\|^2\right] + \frac{1}{\beta_k}\prod_{j=k-t+1}^{k-1}(1 - \beta_j)^2\mathbb{E}\left[\left\|u_k^{(1)}\right\|^2\right] + \beta_k^3 V^2$$

$$(l = 1) \tag{21}$$

*and*

$$\mathbb{E}\left[\left\|y_{k+1}^{(l)} - y_k^{(l)}\right\|^2\right] \le \frac{\beta_k^2}{1 - \beta_k}\mathbb{E}\left[\left\|f^{(l)}\left(y_{k+1}^{(l-1)}\right) - y_{k+1}^{(l)}\right\|^2\right] + \frac{1}{\beta_k}\prod_{j=k-t+1}^{k-1}(1 - \beta_j)^2\mathbb{E}\left[\left\|u_k^{(l)}\right\|^2\right] + \beta_k^3 V^2$$

$$(l \le 2) \tag{22}$$

*Proof.* According to (17), we have

$$\left(y_{k+1}^{(l)} - y_k^{(l)}\right) = \beta_k\left(f_{\xi_{l,k}}^{(l)}\left(y_{k+1}^{(l-1)}\right) - y_k^{(l)}\right) - \prod_{j=k-t+1}^{k-1}(1 - \beta_j)u_k^{(l)}$$

$$= \beta_k\left(f^{(l)}\left(y_{k+1}^{(l-1)}\right) - y_k^{(l)}\right) + \beta_k\left(f_{\xi_{l,k}}^{(l)}(y_{k+1}^{(l-1)}) - f^{(l)}(y_{k+1}^{(l-1)})\right) - \prod_{j=k-t+1}^{k-1}(1 - \beta_j)u_k^{(l)} \tag{23}$$

Taking the square of both sides and using Assumption 4, we have

$$\mathbb{E}\left[\left\|y_{k+1}^{(l)} - y_k^{(l)}\right\|^2\right] = \mathbb{E}\left[\left\|\beta_k\left(f^{(l)}\left(y_{k+1}^{(l-1)}\right) - y_k^{(l)}\right) - \prod_{j=k-t+1}^{k-1}(1 - \beta_j)u_k^{(l)}\right\|^2\right] + \beta_k^3 V^2$$

$$\le \frac{\beta_k^2}{1 - \beta_k}\mathbb{E}\left[\left\|f^{(l)}\left(y_{k+1}^{(l-1)}\right) - y_{k+1}^{(l)}\right\|^2\right] + \frac{1}{\beta_k}\prod_{j=k-t+1}^{k-1}(1 - \beta_j)^2\mathbb{E}\left[\left\|u_k^{(l)}\right\|^2\right] + \beta_k^3 V^2$$

$$\tag{24}$$

$\square$

**Lemma 4.** *The difference between the stochastic gradient and the true gradient is bounded, i.e.,*

$$
\|\mathbb{E}\left[\nabla_k | \mathcal{F}_k\right] - \nabla F\left(\theta_k\right)\| \le \sum_{l=2}^{T+1} \sum_{m=1}^{l-1} A_{m,l} \mathbb{E}\left[\left\|y_{k+1}^{(m)} - f^{(m)}\left(y_{k+1}^{(m-1)}\right)\right\| | \mathcal{F}_k\right]
$$
$$
:= \sum_{l=1}^{T} A_l \mathbb{E}\left[\left\|y_{k+1}^{(l)} - f^{(l)}\left(y_{k+1}^{(l-1)}\right)\right\| | \mathcal{F}_k\right] \tag{25}
$$

*where $A_l := \sum_{m=l+1}^{T} A_{l,m}$.*

This lemma is commonly used in the analysis of SCO algorithms Chen et al. (2020); Yang et al. (2019). Proof can be found in Chen et al. (2020), page 20-23.

**Lemma 5.** *The difference between the composite function and its moving average satisfies*

$$
\sum_{k=0}^{K-1} \sum_{l=1}^{T} \mathbb{E}\left[\left\|y_{k+1}^{(l)} - f^{(l)}\left(y_{k+1}^{(l-1)}\right)\right\|^2\right]
$$
$$
\le \frac{4C_{max}C_1^2 T \alpha^2}{\beta^2} \sum_{k=0}^{K-1} \mathbb{E}\left[\|\nabla F(\theta_k)\|^2\right]
$$
$$
+ 2C_{max}T^2 K\left(\frac{(1-\beta)^{2t-1}}{\beta^3} + \frac{C_{max}(1-\beta)^{2(t-1)}}{\beta^3} + \frac{(1-\beta)^{2t+1}}{T\beta^3}\right)D^2
$$
$$
+ 2C_{max}T^2 K\left(\beta C_{max} + \frac{\beta^2(1-\beta)(1+T)}{T}\right)V^2, \tag{26}
$$

where $C_{max} = max(C_2^2 C_3^2 \cdots C_l^2, C_3^2 C_4^2 \cdots C_l^2, \cdots, C_l^2, 1)$ for $l = 2 \ldots T$.

*Proof.* According to (17), we have

$$
y_{k+1}^{(l)} - f^{(l)}\left(y_{k+1}^{(l-1)}\right)
$$
$$
= (1-\beta_k)\left(y_k^{(l)} - f^{(l)}\left(y_{k+1}^{(l-1)}\right)\right) + \beta_k\left(f_{\xi_{l,k}}^{(l)}\left(y_{k+1}^{(l-1)}\right) - f^{(l)}\left(y_{k+1}^{(l-1)}\right)\right) - \prod_{j=k-t+1}^{k-1}(1-\beta_j)u_k^{(l)}
$$
$$
= (1-\beta_k)\left(y_k^{(l)} - f^{(l)}\left(y_k^{(l-1)}\right)\right) + (1-\beta_k)\left(f^{(l)}\left(y_k^{(l-1)}\right) - f^{(l)}\left(y_{k+1}^{(l-1)}\right)\right)
$$
$$
+ \beta_k\left(f_{\xi_{l,k}}^{(l)}\left(y_{k+1}^{(l-1)}\right) - f^{(l)}\left(y_{k+1}^{(l-1)}\right)\right) - \prod_{j=k-t+1}^{k-1}(1-\beta_j)u_k^{(l)}
$$
$$
= (1-\beta_k)\left(y_k^{(l)} - f^{(l)}\left(y_k^{(l-1)}\right)\right) + \underbrace{f^{(l)}\left(y_k^{(l-1)}\right) - f^{(l)}\left(y_{k+1}^{(l-1)}\right)}_{:=Y_k^{(l)}}
$$
$$
+ \beta_k\left(f_{\xi_{l,k}}^{(l)}\left(y_{k+1}^{(l-1)}\right) - f^{(l)}\left(y_k^{(l-1)}\right)\right) - \prod_{j=k-t+1}^{k-1}(1-\beta_j)u_k^{(l)}. \tag{27}
$$

Taking the square of both sides of (27) and taking the expectation conditioned on $\mathcal{F}_{k,n} := \left\{ \mathcal{F}_k, y_{k+1}^{(1)}, \cdots, y_{k+1}^{(l-1)} \right\}$, we have:

$$\mathbb{E}\left[ \left\| y_{k+1}^{(l)} - f^{(l)}\left( y_{k+1}^{(l-1)} \right) \right\|^2 \right]$$

$$= \mathbb{E}\left[ \left\| (1 - \beta_k)\left( y_k^{(l)} - f^{(l)}\left( y_k^{(l-1)} \right) \right) + Y_k^{(l)} - \prod_{j=k-t+1}^{k-1} (1 - \beta_j) u_k^{(l)} \right\|^2 \right]$$

$$+ \beta_k^2 \mathbb{E}\left[ \left\| \left( f_{\xi_{l,k}}^{(l)}\left( y_{k+1}^{(l-1)} \right) - f^{(l)}\left( y_k^{(l-1)} \right) \right) \right\|^2 \right] \tag{28}$$

$$\overset{(Assumption4)}{\leq} (1 - \beta_k)\,\mathbb{E}\left[ \left\| y_k^{(l)} - f^{(l)}\left( y_k^{(l-1)} \right) \right\|^2 \right] + \frac{1}{\beta_k} \cdot \frac{1}{1 - \beta_k} \mathbb{E}\left[ \left\| Y_k^{(l)} \right\|^2 \right]$$

$$+ \frac{1}{\beta_k^2} \prod_{j=k-t+1}^{k} (1 - \beta_j)^2\,\mathbb{E}\left[ \left\| u_k^{(l)} \right\|^2 \right] + \beta_k^3 V^2.$$

Let us define

$$S_{k+1}^{(l)} = \mathbb{E}\left[ \left\| y_{k+1}^{(l)} - f^{(l)}\left( y_{k+1}^{(l-1)} \right) \right\|^2 \right], \tag{29}$$

Formula (28) can be rewritten as

$$S_{k+1}^{(l)} \leq (1 - \beta_k) S_k^{(l)} + \frac{1}{\beta_k} \cdot \frac{1}{1 - \beta_k} \mathbb{E}\left[ \left\| Y_k^{(l)} \right\|^2 \right]$$

$$+ \frac{1}{\beta_k^2} \prod_{j=k-t+1}^{k} (1 - \beta_j)^2\,\mathbb{E}\left[ \left\| u_k^{(l)} \right\|^2 \right] + \beta_k^3 V^2. \tag{30}$$

Because

$$\mathbb{E}\left[ \left\| Y_k^{(l)} \right\|^2 \right] = \mathbb{E}\left[ \left\| f^{(l)}(y_{k+1}^{(l-1)}) - f^{(l)}(y_k^{(l-1)}) \right\|^2 \right]$$

$$\overset{(Assumption2)}{\leq} C_l^2 \mathbb{E}\left[ \left\| y_{k+1}^{(l-1)} - y_k^{(l-1)} \right\|^2 \right], \tag{31}$$

when $l = 1$,

$$\mathbb{E}\left[ \left\| Y_k^{(1)} \right\|^2 \right] \leq C_1^2 \mathbb{E}\left[ \left\| \theta_{k+1} - \theta_k \right\|^2 \right]$$

$$\leq C_1^2 \mathbb{E}\left[ \left\| \alpha_k \nabla_k + \alpha_k \nabla F(\theta_k) - \alpha_k \nabla F(\theta_k) \right\|^2 \right] \tag{32}$$

$$\leq C_1^2 \cdot \alpha_k^2 \cdot \left( 2\mathbb{E}\left[ \left\| \nabla F(\theta_k) \right\|^2 \right] + 2T \sum_{l=1}^{T} A_l^2 S_{k+1}^{(l)} \right).$$

plugging (32) into (30), we have

$$S_{k+1}^{(1)} \leq (1 - \beta_k) S_k^{(1)} + \frac{1}{\beta_k^2} \prod_{j=k-t+1}^{k-1} (1 - \beta_j)^2 \mathbb{E}\left[ \left\| u_k^{(1)} \right\|^2 \right] + \beta_k^3 V^2$$

$$+ \frac{1}{\beta_k} \cdot \frac{1}{1 - \beta_k} \cdot \mathbb{E}\left[ \left\| Y_k^{(1)} \right\|^2 \right]$$

$$\leq (1 - \beta_k) S_k^{(1)} + \frac{1}{\beta_k^2} \prod_{j=k-t+1}^{k-1} (1 - \beta_j)^2 \mathbb{E}\left[ \left\| u_k^{(1)} \right\|^2 \right] + \beta_k^3 V^2 \tag{33}$$

$$+ \frac{2}{\beta_k} \cdot \frac{1}{1 - \beta_k} \cdot C_1^2 \alpha_k^2 \mathbb{E} \left\| \nabla F(\theta_k) \right\|^2 + \frac{2}{\beta_k} \cdot \frac{1}{1 - \beta_k} \cdot C_1^2 \alpha_k^2 T \sum_{l=1}^{T} A_l^2 S_{k+1}^{(l)}.$$

When $k = 0$, $||u_0^1||^2 = 0$, and

$$S_1^{(1)} \leq (1 - \beta_0)S_0^{(1)} + \beta_0 \underbrace{(\beta_0^2 V^2 + \frac{2}{\beta_0^2} \cdot \frac{1}{1 - \beta_0} C_1^2 \alpha_0^2 \|\nabla F(\theta_0)\|^2 + \frac{2}{\beta_0^2} \cdot \frac{1}{1 - \beta_0} C_1^2 \alpha_0^2 T \sum_{l=1}^{T} S_1^{(l)})}_{:=\Delta_0^{(1)}}$$

$$= (1 - \beta_0)S_0^{(1)} + \beta_0 \Delta_0^{(1)}.$$

(34)

Similarly, we have

$$S_2^{(1)} \leq (1 - \beta_1)S_1^{(1)} + \beta_1 \Delta_1^{(1)}$$

$$\vdots$$

(35)

$$S_{k+1}^{(1)} \leq (1 - \beta_k)S_k^{(1)} + \beta_k \Delta_k^{(1)}.$$

Here, for $i > t$,

$$\Delta_i^{(1)} = \frac{1}{\beta_i^3} \prod_{j=i-t-1}^{i-1} (1 - \beta_j)^2 \mathbb{E}\left[\left\|u_i^{(1)}\right\|^2\right] + \beta_i^2 V^2$$

$$+ \frac{2}{\beta_i^2} \cdot \frac{1}{1 - \beta_i} C_1^2 \alpha_i^2 \mathbb{E}\left[\|\nabla F(\theta_i)\|^2\right]$$

$$+ \frac{2}{\beta_i^2} \cdot \frac{1}{1 - \beta_i} C_1^2 \alpha_i^2 T \sum_{l=1}^{T} S_{i+1}^{(l)};$$

(36)

for $0 < i \leq t$,

$$\Delta_i^{(1)} = \beta_i^2 V^2 + \frac{2}{\beta_i^2} \cdot \frac{1}{1 - \beta_i} C_1^2 \alpha_i^2 \mathbb{E}\left[\|\nabla F(\theta_i)\|^2\right]$$

$$+ \frac{2}{\beta_i^2} \cdot \frac{1}{1 - \beta_i} C_1^2 \alpha_i^2 T \sum_{l=1}^{T} S_{i+1}^{(l)}.$$

(37)

Because $S_0^{(1)} = 0$, we can recursively write:

$$S_1^{(1)} \leq \beta_0 \Delta_0^{(1)}$$

$$S_2^{(1)} \leq (1 - \beta_1)\beta_0 \Delta_0^{(1)} + \beta_1 \Delta_1^{(1)}$$

$$S_3^{(1)} \leq (1 - \beta_2)(1 - \beta_1)\beta_0 \Delta_0^{(1)} + (1 - \beta_2)\beta_1 \Delta_1^{(1)} + \beta_2 \Delta_2^{(1)}$$

$$\vdots$$

$$S_{k+1}^{(1)} \leq (1 - \beta_k) \cdots (1 - \beta_1)\beta_0 \Delta_0^{(1)} + (1 - \beta_k) \cdots (1 - \beta_2)\beta_1 \Delta_1^{(1)} + \cdots$$

$$+ (1 - \beta_k)\beta_{k-1}\Delta_{k-1}^{(1)} + \beta_k \Delta_k^{(1)}$$

$$= \sum_{i=1}^{k} w_{i,k} \Delta_i^{(1)},$$

(38)

where $w_{i,k} = (1 - \beta_k) \cdots (1 - \beta_{i+1})\beta_i$.

For $l > 1$, (30) can be rewritten as

$$S_{k+1}^{(l)} \leq (1 - \beta_k) S_k^{(l)} + \frac{C_l^2}{\beta_k (1 - \beta_k)} \mathbb{E}\left[\left\|y_{k+1}^{(l-1)} - y_k^{(l-1)}\right\|^2\right] + \frac{1}{\beta_k^2} \prod_{j=k-t+1}^{k-1} (1 - \beta_j)^2 \mathbb{E}\left[\left\|u_k^{(l)}\right\|^2\right] + \beta_k^3 V^2$$

$$\overset{\text{(Lemma 3)}}{\leq} (1 - \beta_k) S_k^{(l)} + \frac{\beta_k C_l^2}{(1 - \beta_k)^2} S_{k+1}^{(l-1)} + \beta_k Z_k^{(l)},$$

(39)

where

$$Z_k^{(l)} = \frac{1}{\beta_k^3} \prod_{j=k-t+1}^{k-1} (1-\beta_j)^2 \, \mathbb{E}\left[\left\|u_k^{(l)}\right\|^2\right] + \frac{C_l^2}{\beta_k^3(1-\beta_k)} \prod_{j=k-t+1}^{k-1} (1-\beta_j)^2 \, \mathbb{E}\left[\left\|u_k^{(l-1)}\right\|^2\right] + \beta_k^2 V^2 + \frac{\beta_k C_l^2}{1-\beta_k} V^2.$$

(40)

It follows that

$$\begin{aligned}
S_{k+1}^{(2)} \leq &(1-\beta_k)S_k^{(2)} + \frac{\beta_k C_2^2}{(1-\beta_k)^2} S_{k+1}^{(1)} + \beta_k Z_k^{(2)} \\
\overset{According to (38)}{\leq} &(1-\beta_k)S_k^{(2)} + \frac{\beta_k C_2^2}{(1-\beta_k)^2} \sum_{i=0}^{k} w_{i,k}\Delta_i^{(1)} + \beta_k Z_k^{(2)} \\
= &(1-\beta_k)S_k^{(2)} + \beta_k \left( \frac{C_2^2}{(1-\beta_k)^2} \sum_{i=0}^{k} w_{i,k}\Delta_i^{(1)} + Z_k^{(2)} \right)
\end{aligned}$$

(41)

Following the same steps as (38), by recursively plugging $S_k^{(2)}$ into $S_{k+1}^{(2)}$, we can obtain

$$\begin{aligned}
S_{k+1}^{(2)} \leq &\sum_{j=0}^{k} w_{j,k} \left( \frac{C_2^2}{(1-\beta_j)^2} \sum_{i=0}^{j} w_{i,k}\Delta_i^{(1)} + Z_j^{(2)} \right) \\
= &\sum_{j=0}^{k} \frac{C_2^2 w_{j,k}}{(1-\beta_j)^2} \sum_{i=0}^{j} w_{i,k}\Delta_i^{(1)} + \sum_{j=0}^{k} w_{j,k} Z_j^{(2)} \\
= &\sum_{i=0}^{k} \left( \sum_{j=i}^{k} \frac{C_2^2 w_{j,k}}{(1-\beta_j)^2} \right) w_{i,k}\Delta_i^{(1)} + \sum_{i=0}^{k} w_{i,k} Z_i^{(2)}
\end{aligned}$$

(42)

According to (38), it is easy to find $\sum_{j=i}^{k} w_{j,k} < 1$. Thus, we have

$$\sum_{j=i}^{k} \frac{C_2^2 w_{j,k}}{(1-\beta_j)^2} \leq \sum_{j=i}^{k} \frac{C_2^2 w_{j,k}}{(1-\beta_i)^2} \leq \frac{C_2^2}{(1-\beta_i)^2} \sum_{j=i}^{k} w_{j,k} \leq \frac{C_2^2}{(1-\beta_i)^2}.$$

(43)

Combining (43) and (42), we have

$$\begin{aligned}
S_{k+1}^{(2)} \leq &\sum_{i=0}^{k} \left( \frac{C_2^2}{(1-\beta_i)^2} w_{i,k}\Delta_i^{(1)} + w_{i,k} Z_i^{(2)} \right) \\
\leq &\sum_{i=0}^{k} w_{i,k} \left( \frac{C_2^2}{(1-\beta_i)^2} \Delta_i^{(1)} + Z_i^{(2)} \right) \\
= &\sum_{i=0}^{k} w_{i,k}\Delta_i^{(2)}
\end{aligned}$$

(44)

Similarly, for any $l > 1$, we have

$$S_{k+1}^{(l)} \leq \sum_{i=0}^{k} w_{i,k}\Delta_i^{(l)}$$

(45)

where

$$\Delta_i^{(2)} = \frac{C_2^2}{(1-\beta_i)^2}\Delta_i^{(1)} + Z_i^{(2)}$$

$$\Delta_i^{(3)} = \frac{C_3^2}{(1-\beta_i)^2}\Delta_i^{(2)} + Z_i^{(3)}$$

$$= \frac{C_2^2 C_3^2}{(1-\beta_i)^4}\Delta_i^{(1)} + \frac{C_3^2}{(1-\beta_i)^2}Z_i^{(2)} + Z_i^{(3)} \quad (46)$$

$$\Delta_i^{(l)} = \frac{C_2^2 C_3^2 \cdots C_l^2}{(1-\beta_i)^{2(l-1)}}\Delta_i^{(1)} + \frac{C_3^2 \cdots C_l^2}{(1-\beta_i)^{2(l-2)}}Z_i^{(2)} + \frac{C_4^2 \cdots C_l^2}{(1-\beta_i)^{2(l-3)}}Z_i^{(3)} + \cdots + Z_i^{(l)}$$

Let $C_{max} = max(C_2^2 C_3^2 \cdots C_l^2, C_3^2 C_4^2 \cdots C_l^2, \cdots, C_l^2, 1), \forall l \in [1, T]$ and $Z_i^{max}$ defined as follows:

$$Z_i^{max} = \frac{1}{\beta_i^3}\prod_{j=i-t+1}^{i-1}(1-\beta_j)^2 D^2 + \frac{C_{max}}{\beta_i^3(1-\beta_i)}\prod_{j=i-t+1}^{i-1}(1-\beta_j)^2 D^2 + \beta_i^2 V^2 + \frac{\beta_i C_{max}}{1-\beta_i}V^2 \quad (47)$$

Plugging (46) into (45), we can obtain

$$S_{k+1}^{(l)} \le \sum_{i=0}^{k}\frac{C_{max}}{(1-\beta_i)^{2(T-1)}}w_{i,k}\Delta_i^{(1)} + \sum_{i=0}^{k}w_{i,k}\sum_{j=2}^{l}\frac{C_{max}}{(1-\beta_i)^{2(T-1)}}Z_i^{(l)}. \quad (48)$$

Summing (48) from $l = 1$ to $T$, setting $\beta_k = \beta$ and $\alpha_k = \alpha$, then substituting $\Delta_k^{(1)}$ we have:

$$\sum_{k=0}^{K-1}\sum_{l=1}^{T}S_{k+1}^{(l)} \le \sum_{k=0}^{K-1}\sum_{i=0}^{k}\sum_{l=1}^{T}\frac{C_{max}}{(1-\beta)^{2(T-1)}}w_{i,k}\Delta_i^{(1)} + \sum_{k=0}^{K-1}\sum_{i=0}^{k}w_{i,k}\sum_{l=1}^{T}\sum_{j=2}^{l}\frac{C_{max}}{(1-\beta)^{2(T-1)}}Z_i^{(l)}$$

$$\le \sum_{k=0}^{K-1}\sum_{i=k}^{K-1}\frac{C_{max}T}{(1-\beta)^{2(T-1)}}w_{i,K-1}\Delta_k^{(1)} + \sum_{k=0}^{K-1}\sum_{i=k}^{K-1}\frac{C_{max}T^2}{(1-\beta)^{2(T-1)}}w_{i,K-1}Z_i^{max}$$

$$\le \sum_{k=0}^{K-1}\frac{C_{max}T}{(1-\beta)^{2(T-1)}}\Delta_k^{(1)} + \sum_{k=0}^{K-1}\sum_{i=k}^{K-1}\frac{C_{max}T^2}{(1-\beta)^{2(T-1)}}w_{i,K-1}Z_i^{max}$$

$$\le \frac{C_{max}T}{(1-\beta)^{2(T-1)}}\sum_{k=0}^{K-1}\left(\beta^2 V^2 + \frac{2C_1^2\alpha^2}{\beta^2(1-\beta)}\mathbb{E}\left[\|\nabla F(\theta_k)\|^2\right] + \frac{2C_1^2\alpha^2 T}{\beta^2(1-\beta)}\sum_{l=1}^{T}S_{k+1}^{(l)}\right)$$

$$+ \frac{C_{max}TD^2K}{\beta^3(1-\beta)^{2(T-1-t)}} + \frac{C_{max}T^2KZ^{max}}{(1-\beta)^{2(T-1)}}. \quad (49)$$

It follows that

$$\sum_{k=0}^{K-1}\sum_{l=1}^{T}S_{k+1}^{(l)} \le \frac{C_{max}T\beta^2(1-\beta)}{\beta^2(1-\beta)^{2T-1} - 2C_1^2 C_{max}\alpha^2 T^2}\sum_{k=0}^{K-1}\left(\beta^2 V^2 + \frac{2C_1^2\alpha^2}{\beta^2(1-\beta)}\mathbb{E}\left[\|\nabla F(\theta_k)\|^2\right]\right)$$

$$+ \frac{C_{max}TD^2K(1-\beta)^{2t+1}}{\beta^3(1-\beta)^{2T-1} - 2C_1^2 C_{max}T^2\alpha^2\beta} + \frac{C_{max}T^2\beta^2(1-\beta)KZ^{max}}{\beta^2(1-\beta)^{2T-1} - 2C_1^2 C_{max}\alpha^2 T^2} \quad (50)$$

If we set $\beta \leq 1 - \sqrt[2T-1]{\frac{1}{2}}$ and $\alpha \leq \frac{\beta}{2C_1 T}\sqrt{\frac{2(1-\beta)^{2T-1}-1}{C_{max}}}$, we can ensure $\beta^2(1-\beta)^{2T-1} - 2C_1^2 C_{max}\alpha^2 T^2 \geq \beta^2/2$, and (50) can be simplified as

$$
\begin{aligned}
\sum_{k=0}^{K-1}\sum_{l=1}^{T} S_{k+1}^{(l)} \leq & 2C_{max}TK\beta^2(1-\beta)^2 V^2 + \frac{4C_{max}C_1^2 T\alpha^2}{\beta^2}\sum_{k=0}^{K-1}\mathbb{E}\left[\|\nabla F(\theta_k)\|^2\right] \\
& + \frac{2C_{max}TK(1-\beta)^{2t+1}}{\beta^3}D^2 + 2C_{max}T^2 K(1-\beta)Z^{max} \\
\leq & 2C_{max}TK\beta^2(1-\beta)^2 V^2 + \frac{4C_{max}C_1^2 T\alpha^2}{\beta^2}\sum_{k=0}^{K-1}\mathbb{E}\left[\|\nabla F(\theta_k)\|^2\right] \\
& + \frac{2C_{max}TK(1-\beta)^{2t+1}}{\beta^3}D^2 + 2C_{max}T^2 K(1-\beta)\left(\beta^2 + \frac{\beta C_{max}}{1-\beta}\right)V^2 \\
& + \frac{2C_{max}T^2 K(1-\beta)^{2t-1}}{\beta^3}(1+\frac{C_{max}}{1-\beta})D^2 \\
\leq & \frac{4C_{max}C_1^2 T\alpha^2}{\beta^2}\sum_{k=0}^{K-1}\mathbb{E}\left[\|\nabla F(\theta_k)\|^2\right] \\
& + 2C_{max}T^2 K\left(\frac{(1-\beta)^{2t-1}}{\beta^3} + \frac{C_{max}(1-\beta)^{2(t-1)}}{\beta^3} + \frac{(1-\beta)^{2t+1}}{T\beta^3}\right)D^2 \\
& + 2C_{max}T^2 K\left(\beta C_{max} + \frac{\beta^2(1-\beta)(1+T)}{T}\right)V^2
\end{aligned}
$$
(51)

$\square$

### 8.1.2 Remaining steps towards Theorem 1.

Using the smoothness of $F(\theta_k)$, we have

$$
\begin{aligned}
F(\theta_{k+1}) \leq & F(\theta_k) + \langle \nabla F(\theta_k), \theta_{k+1} - \theta_k \rangle + \frac{L}{2}\|\theta_{k+1} - \theta_k\|^2 \\
= & F(\theta_k) - \alpha_k\langle\nabla F(\theta_k), g_{k+1}\rangle + \frac{L}{2}\alpha_k^2\|g_{k+1}\|^2 \\
= & F(\theta_k) - \alpha_k\mathbb{E}\left[\|\nabla F(\theta_k)\|^2\right] + \frac{L\alpha_k^2}{2}\|g_{k+1}\|^2 + \alpha_k\langle\nabla F(\theta_k), \nabla F(\theta_k) - g_{k+1}\rangle \\
= & F(\theta_k) - \alpha_k\mathbb{E}\left[\|\nabla F(\theta_k)\|^2\right] + \frac{L\alpha_k^2}{2}\|g_{k+1}\|^2 + \alpha_k\langle\nabla F(\theta_k), \nabla F(\theta_k) - \nabla_k\rangle + \alpha_k\langle\nabla F(\theta_k), \nabla_k - g_{k+1}\rangle.
\end{aligned}
$$
(52)

Conditioning on $\mathcal{F}_k$ and taking expectation of both sides w.r.t. $\xi_k$, we have

$$
\begin{aligned}
& \mathbb{E}\left[F\left(\theta_{k+1}\right)|\mathcal{F}_k\right] \\
\leq & F(\theta_k) - \alpha_k\mathbb{E}\left[\|\nabla F(\theta_k)\|^2\right] + \frac{L}{2}\mathbb{E}\left[\|\theta_{k+1} - \theta_k\|^2 \mid \mathcal{F}_k\right] + \alpha_k\langle\nabla F(\theta_k), \mathbb{E}\left[\nabla F(\theta_k) - \nabla_k|\mathcal{F}_k\right]\rangle \\
\leq & F(\theta_k) - \alpha_k\mathbb{E}\left[\|\nabla F(\theta_k)\|^2\right] + \frac{L}{2}C_1^2\cdots C_T^2\alpha_k^2 + \alpha_k\mathbb{E}\left[\|\nabla F(\theta_k)\|\right]\|\mathbb{E}\left[\nabla_k|\mathcal{F}_k\right] - \nabla F(\theta_k)\| \\
\overset{(\text{Lemma 4})}{\leq} & F(\theta_k) - \alpha_k\mathbb{E}\left[\|\nabla F(\theta_k)\|^2\right] + \frac{L}{2}C_1^2\cdots C_T^2\alpha_k^2 + \alpha_k\sum_{l=1}^{T}\left(A_l\|\nabla F(\theta_k)\|\mathbb{E}\left[\left\|y_{k+1}^{(l)} - f^{(l)}(y_{k+1}^{(l-1)})|\mathcal{F}_k\right\|\right]\right） \\
\leq & F(\theta_k) - \alpha_k\left(1 - \frac{\alpha_k}{4\beta_k}\sum_{l=1}^{T}A_l^2\right)\mathbb{E}\left[\|\nabla F(\theta_k)\|^2\right] + \beta_k\sum_{l=1}^{T}\mathbb{E}\left[\left\|y_{k+1}^{(l)} - f^{(l)}(y_{k+1}^{(l-1)})\right\|^2 |\mathcal{F}_k\right] + \frac{L}{2}C_1^2\cdots C_T^2\alpha_k^2.
\end{aligned}
$$
(53)

Summing from $k = 0$ to $K - 1$, we have

$$
\sum_{k=0}^{K-1} \mathbb{E}\left[F(\theta_{k+1}|\mathcal{F}_k)\right] \leq \sum_{k=0}^{K-1} F(\theta_k) - \sum_{k=0}^{K-1} \alpha_k (1 - \frac{\alpha_k}{4\beta_k} \sum_{l=1}^{T} A_l^2) \mathbb{E}\left[\|\nabla F(\theta_k)\|\right]^2
$$
$$
+ \sum_{k=0}^{K-1} \beta_k \sum_{l=1}^{T} S_{k+1}^{(l)} + \sum_{k=0}^{K-1} \frac{L}{2} C_1^2 \cdots C_T^2 \alpha_k^2.
$$
(54)

Substitute (50) into the above equation, we have

$$
\sum_{k=0}^{K-1} \mathbb{E}\left[F(\theta_{k+1}|\mathcal{F}_k)\right] \leq \sum_{k=0}^{K-1} F(\theta_k) - \alpha(1 - \frac{\alpha}{4\beta} \sum_{l=1}^{T} A_l^2) \sum_{k=0}^{K-1} \mathbb{E}\left[\|\nabla F(\theta_k)\|\right]^2
$$
$$
+ \frac{4 C_{max} C_1^2 T \alpha^2}{\beta} \sum_{k=0}^{K-1} \mathbb{E}\left[\|\nabla F(\theta_k)\|^2\right] + \frac{LK}{2} C_1^2 \cdots C_T^2 \alpha^2
$$
$$
+ 2 C_{max} T^2 K \left( \frac{(1-\beta)^{2t-1}}{\beta^2} + \frac{C_{max}(1-\beta)^{2(t-1)}}{\beta^2} + \frac{(1-\beta)^{2t+1}}{T\beta^2} \right) D^2
$$
$$
+ 2 C_{max} T^2 K \left( \beta^2 C_{max} + \frac{\beta^3(1-\beta)(1+T)}{T} \right) V^2.
$$
(55)

If we set $\alpha \leq \frac{2\beta}{\sum_{l=1}^{T} A_l^2 + 16 C_{max} C_1^2 T}$, we can ensure $1 - \frac{\alpha}{4\beta} \sum_{l=1}^{T} A_l^2 - \frac{4 C_{max} C_1^2 T \alpha}{\beta} \geq \frac{1}{2}$, and thus,

$$
\sum_{k=0}^{K-1} \mathbb{E}\left[F(\theta_{k+1}|\mathcal{F}_k)\right] \leq \sum_{k=0}^{K-1} F(\theta_k) + \frac{\alpha}{2} \sum_{k=0}^{K-1} \mathbb{E}\left[\|\nabla F(\theta_k)\|\right]^2 + \frac{LK}{2} C_1^2 \cdots C_T^2 \alpha^2
$$
$$
+ 2 C_{max} T^2 K \left( \frac{(1-\beta)^{2t-1}}{\beta^2} + \frac{C_{max}(1-\beta)^{2(t-1)}}{\beta^2} + \frac{(1-\beta)^{2t+1}}{T\beta^2} \right) D^2
$$
$$
+ 2 C_{max} T^2 K \left( \beta^2 C_{max} + \frac{\beta^3(1-\beta)(1+T)}{T} \right) V^2.
$$
(56)

Combining with the condition for $\alpha$ and $\beta$ in (50), if we set

$$
\beta \leq 1 - 2^{(-1/(2T-1))}
$$
$$
\alpha = \beta \cdot \min \left( \frac{1}{2 C_1 T} \sqrt{\frac{2(1-\beta)^{2T-1} - 1}{C_{max}}}, \frac{2}{\sum_{l=1}^{T} A_l^2 + 16 C_{max} C_1^2 T} \right),
$$
(57)

we have

$$
\frac{1}{K} \sum_{k=0}^{K-1} \mathbb{E}\left[\|\nabla F(\theta_k)\|\right]^2 \leq \frac{2(F(\theta_0) - F(\theta^*))}{K\alpha} + L C_1^2 \cdots C_T^2 \alpha
$$
$$
+ 4 C_{max} T^2 \left( \frac{(1-\beta)^{2t-1}}{\alpha\beta^2} + \frac{C_{max}(1-\beta)^{2(t-1)}}{\alpha\beta^2} + \frac{(1-\beta)^{2t+1}}{T\alpha\beta^2} \right) D^2
$$
$$
+ 4 C_{max} T^2 \left( \frac{\beta^2 C_{max}}{\alpha} + \frac{\beta^3(1-\beta)(1+T)}{\alpha T} \right) V^2
$$
$$
\leq O(\beta) + 4 C_{max} T^2 \left( \frac{1 + C_{max})(1-\beta)^{2(t-1)}}{\alpha\beta^2} + \frac{(1-\beta)^{2t+1}}{T\alpha\beta^2} \right) D^2
$$
(58)

This completes the proof of Theorem 1. If we can set $t$ such that $(1-\beta)^{2(t-1)} = O(\beta^4)$, the algorithm achieves $O(1/\sqrt{K})$ convergence rate.

## 8.2 PROOF TO THEOREM 2

The SCSC algorithm with our sparse moving average can be written as

$$y_{k+1}^{(1)} = (1 - \beta_k)y_k^{(1)} + f_{\xi_{1,k}}^{(1)}(\theta_k) - (1 - \beta_k)f_{\xi_{1,k}}^{(1)}(\theta_{k-1}) - \prod_{j=k-t}^{k-1}(1 - \beta_j)u_k^{(1)}, \qquad (59)$$

$$y_{k+1}^{(l)} = (1 - \beta_k)y_k^{(l)} + f_{\xi_{l,k}}^{(l)}(y_{k+1}^{(l-1)}) - (1 - \beta_k)f_{\xi_{l,k}}^{(l)}(y_k^{(l-1)}) - \prod_{j=k-t}^{k-1}(1 - \beta_j)u_k^{(l)}, 2 \le l \le T, \quad (60)$$

$$\theta_{k+1} = \theta_k - \alpha_k \nabla_k, \qquad (61)$$

with

$$u_k^{(l)} = P(\xi_{1,k-t}/(\xi_{1,k-t+1} \cup \cdots \cup \xi_{1,k}))y_{k+t-1}^{(l)} \qquad (62)$$

and

$$\nabla_k = \nabla f_{\xi_{1,k}}^{(1)}(\theta_k) \cdots \nabla f_{\xi_{N,k}}^{(T)}(y_{k+1}^{(T-1)})\nabla f_{\xi_{T+1,k}}^{(T+1)}(y_{k+1}^{(T)}). \qquad (63)$$

### 8.2.1 SUPPORTING LEMMA

**Lemma 6.** *The change of $y_k^{(l)}$ in every iteration is bounded, i.e.,*

$$\begin{aligned}
\mathbb{E}\left[\left\|y_{k+1}^{(l)} - y_k^{(l)}\right\|^2\right] \le& 3\left(\frac{\beta_k}{1-\beta_k}\right)^2 \mathbb{E}\left[\left\|y_{k+1}^{(l)} - f^{(l)}\left(y_{k+1}^{(l-1)}\right)\right\|^2\right] \\
&+ \left(\frac{\beta_k}{1-\beta_k}\right)^2 V^2 + 3C_{l-1}^2 \mathbb{E}\left[\left\|y_{k+1}^{(l-1)} - y_k^{(l-1)}\right\|^2\right] \\
&+ 3\prod_{j=k-t}^{k-1}(1-\beta_j)^2 \mathbb{E}\left[\left\|u_k^{(l)}\right\|^2\right]
\end{aligned} \qquad (64)$$

*Proof.* According to (60), we have

$$\begin{aligned}
(1 - \beta_k)\left(y_{k+1}^{(l)} - y_k^{(l)}\right) =& \beta_k\left(f_{\xi_{l,k}}^{(l)}\left(y_{k+1}^{(l-1)}\right) - y_{k+1}^{(l)}\right) \\
&+ (1-\beta_k)\left(f_{\xi_{l,k}}^{(l)}\left(y_{k+1}^{(l-1)}\right) - f_{\xi_{l,k}}^{(l)}\left(y_k^{(l-1)}\right)\right) - \prod_{j=k-t}^{k-1}(1-\beta_j)u_k^{(l)} \\
=& \beta_k\left(f^{(l)}\left(y_{k+1}^{(l-1)}\right) - y_{k+1}^{(l)}\right) + \beta_k\left(f_{\xi_{l,k}}^{(l)}\left(y_{k+1}^{(l-1)}\right) - f^{(l)}\left(y_{k+1}^{(l-1)}\right)\right) \\
&+ (1-\beta_k)\left(f_{\xi_{l,k}}^{(l)}\left(y_{k+1}^{(l-1)}\right) - f_{\xi_{l,k}}^{(l)}\left(y_k^{(l-1)}\right)\right) - \prod_{j=k-t}^{k-1}(1-\beta_j)u_k^{(l)}
\end{aligned} \qquad (65)$$

Taking the square of both sides of (65) gives us (64). $\qquad \square$

**Lemma 7.** *The difference between the composite function and its moving average is bounded, i.e.,*

$$\begin{aligned}
\mathbb{E}\left[\left\|y_{k+1}^{(l)} - f^{(l)}\left(y_{k+1}^{(l-1)}\right)\right\|^2\right] \le& (1 - \beta_k)\mathbb{E}\left[\left\|y_k^{(l)} - f^{(l)}\left(y_k^{(l-1)}\right)\right\|^2\right] \\
&+ 4(1-\beta_k)^2 C_l^2 \mathbb{E}\left[\left\|y_k^{(l-1)} - y_{k+1}^{(l-1)}\right\|^2\right] + 2\beta_k^2 V^2 \\
&+ \frac{1}{\beta_k}\prod_{j=k-t}^{k-1}(1-\beta_j)^2 \mathbb{E}\left[\|u_k^{(l)}\|^2\right]
\end{aligned} \qquad (66)$$

*Proof.*

$$
y_{k+1}^{(l)} - f^{(l)}\left(y_{k+1}^{(l-1)}\right)
$$
$$
= (1 - \beta_k)\left(y_k^{(l)} - f^{(l)}\left(y_k^{(l-1)}\right)\right) + (1 - \beta_k)\underbrace{\left(f^{(l)}\left(y_k^{(l-1)}\right) - f^{(l)}\left(y_{k+1}^{(l-1)}\right)\right)}_{:=T_1}
$$
$$
+ \beta_k \underbrace{\left(f_{\xi_{l,k}}^{(l)}\left(y_{k+1}^{(l-1)}\right) - f^{(l)}\left(y_{k+1}^{(l-1)}\right)\right)}_{:=T_2} \tag{67}
$$
$$
+ (1 - \beta_k)\underbrace{\left(f_{\xi_{l,k}}^{(l)}\left(y_{k+1}^{(l-1)}\right) - f_{\xi_{l,k}}^{(l)}\left(y_k^{(l-1)}\right)\right)}_{:=T_3} + \prod_{j=k-t}^{k-1}(1 - \beta_j)\, u_k^{(l)}
$$

Conditioned on $\mathcal{F}_k$, taking expectation over $\xi_{l,k}$, we have:

$$
E\left[(1 - \beta_k)\,T_1 + \beta_k T_2 + (1 - \beta_k)\,T_3\right]
$$
$$
= E\left[(1 - \beta_k)\,f^{(l)}\left(y_k^{(l-1)}\right) - f^{(l)}\left(y_{k+1}^{(l-1)}\right) + f_{\xi_{l,k}}^{(l)}\left(y_{k+1}^{(l)}\right) - (1 - \beta_k)\,f_{\xi_{l,k}}^{(l)}\left(y_k^{(l)}\right)\right]
$$
$$
= 0
$$

Taking the square of both sides of (67) and taking the expectation conditioned on $\mathcal{F}_{k,n} := \left\{\mathcal{F}_k, y_{k+1}^{(1)}, \cdots, y_{k+1}^{(l-1)}\right\}$, we have:

$$
\mathbb{E}\left[\left\|y_{k+1}^{(l)} - f^{(l)}\left(y_{k+1}^{(l-1)}\right)\right\|^2\right]
$$
$$
= \mathbb{E}\left[\left\|(1 - \beta_k)\left(y_k^{(l)} - f^{(l)}\left(y_k^{(l-1)}\right)\right) - \prod_{j=k-t}^{k-1}(1 - \beta_j)\, u_k^{(l)}\right\|^2\right]
$$
$$
+ \mathbb{E}\left[\left\|(1 - \beta_k)\,T_1 + \beta_k T_2 + (1 - \beta_k)\,T_3\right\|^2\right]
$$
$$
\leq (1 - \beta_k)\,\mathbb{E}\left[\left\|y_k^{(l)} - f^{(l)}\left(y_k^{(l-1)}\right)\right\|^2\right] + \frac{1}{\beta_k}\prod_{j=k-t}^{k-1}(1 - \beta_j)^2\,\mathbb{E}\left[\|u_k^{(l)}\|^2\right]
$$
$$
+ 2\mathbb{E}\left[\left\|(1 - \beta_k)\,T_1 + \beta_k T_2\right\|^2 |\mathcal{F}_{k,n}\right] + 2(1 - \beta_k)^2\,\mathbb{E}\left[\|T_3\|^2 |\mathcal{F}_{k,n}\right]
$$
$$
\leq (1 - \beta_k)\,\mathbb{E}\left[\left\|y_k^{(l)} - f^{(l)}\left(y_k^{(l-1)}\right)\right\|^2\right] + 2(1 - \beta_k)^2\,\mathbb{E}\left[\|T_1\|^2 |\mathcal{F}_{k,n}\right] + 2\beta_k^2\mathbb{E}\left[\|T_2\|^2 |\mathcal{F}_{k,n}\right]
$$
$$
+ 2(1 - \beta_k)^2\,\mathbb{E}\left[\|T_3\|^2 |\mathcal{F}_{k,n}\right] + \frac{1}{\beta_k}\prod_{j=k-t}^{k-1}(1 - \beta_j)^2\,\mathbb{E}\left[\|u_k^{(l)}\|^2\right]
$$
$$
\leq (1 - \beta_k)\,\mathbb{E}\left[\left\|y_k^{(l)} - f^{(l)}\left(y_k^{(l-1)}\right)\right\|^2\right] + 2(1 - \beta_k)^2\,\mathbb{E}\left[\left\|f^{(l)}\left(y_k^{(l-1)}\right) - f^{(l)}\left(y_{k+1}^{(l-1)}\right)\right\|^2\right]
$$
$$
+ 2(1 - \beta_k)^2\,\mathbb{E}\left[\left\|f_{\xi_{l,k}}^{(l)}\left(y_k^{(l-1)}\right) - f_{\xi_{l,k}}^{(l)}\left(y_{k+1}^{(l-1)}\right)\right\|^2\right] + 2\beta_k^2 V^2
$$
$$
+ \frac{1}{\beta_k}\prod_{j=k-t}^{k-1}(1 - \beta_j)^2\,\mathbb{E}\left[\|u_k^{(l)}\|^2\right]
$$

$$
\tag{68}
$$

$$\leq (1 - \beta_k) \mathbb{E}\left[\left\|y_k^{(l)} - f^{(l)}\left(y_k^{(l-1)}\right)\right\|^2\right]$$

$$+ 4(1 - \beta_k)^2 C_l^2 \mathbb{E}\left[\left\|y_k^{(l-1)} - y_{k+1}^{(l-1)}\right\|^2\right] + 2\beta_k^2 V^2 \tag{69}$$

$$+ \frac{1}{\beta_k} \prod_{j=k-t}^{k-1} (1 - \beta_j)^2 \mathbb{E}\left[\|u_k^{(l)}\|^2\right]$$

$\square$

### 8.2.2 Remaining steps towards Theorem 2.

Using the smoothness of $F(\theta_k)$, we have

$$F(\theta_{k+1}) \leq F(\theta_k) + \langle \nabla F(\theta_k), \theta_{k+1} - \theta_k \rangle + \frac{L}{2} \|\theta_{k+1} - \theta_k\|^2$$

$$= F(\theta_k) - \alpha_k \langle \nabla F(\theta_k), \nabla_k \rangle + \frac{L}{2} \|\theta_{k+1} - \theta_k\|^2$$

$$= F(\theta_k) - \alpha_k \mathbb{E}[\|\nabla F(\theta_k)\|^2] + \frac{L}{2} \|\theta_{k+1} - \theta_k\|^2 + \alpha_k \langle \nabla F(\theta_k), \nabla F(\theta_k) - \nabla_k \rangle .$$

$$\tag{70}$$

Conditioning on $\mathcal{F}_k$ and taking expectation of both sides w.r.t. $\xi_k$, we have

$$\mathbb{E}\left[F\left(\theta_{k+1}\right) | \mathcal{F}_k\right]$$

$$\leq F(\theta_k) - \alpha_k \mathbb{E}\left[\|\nabla F(\theta_k)\|^2\right] + \frac{L}{2}\mathbb{E}\left[\|\theta_{k+1} - \theta_k\|^2 \mid \mathcal{F}_k\right] + \alpha_k \langle \nabla F(\theta_k), \mathbb{E}\left[\nabla F(\theta_k) - \nabla_k | \mathcal{F}_k\right]\rangle$$

$$\leq F(\theta_k) - \alpha_k \mathbb{E}\left[\|\nabla F(\theta_k)\|^2\right] + \frac{L}{2}C_1^2 \cdots C_{T+1}^2 \alpha_k^2 + \alpha_k \mathbb{E}\left[\|\nabla F(\theta_k)\|\right] \|\mathbb{E}\left[\nabla_k | \mathcal{F}_k\right] - \nabla F(\theta_k)\|$$

$$\overset{\text{(Lemma 4)}}{\leq} F(\theta_k) - \alpha_k \mathbb{E}\left[\|\nabla F(\theta_k)\|^2\right] + \frac{L}{2}C_1^2 \cdots C_{T+1}^2 \alpha_k^2 + \alpha_k \sum_{l=1}^{T} A_l \mathbb{E}\left[\|\nabla F(\theta_k)\|\right] \mathbb{E}\left[\left\|y_{k+1}^{(T)} - f^{(T)}(y_{k+1}^{(T-1)})\right\| | \mathcal{F}_k\right]$$

$$\leq F(\theta_k) - \alpha_k \left(1 - \frac{\alpha_k}{2\beta_k}\sum_{l=1}^{T} A_l^2\right) \mathbb{E}\left[\|\nabla F(\theta_k)\|^2\right] + \frac{\beta_k}{2}\sum_{l=1}^{T}\mathbb{E}\left[\left\|y_{k+1}^{(T)} - f^{(T)}(y_{k+1}^{(T-1)})\right\|^2 | \mathcal{F}_k\right]$$

$$+ \frac{L}{2}C_1^2 \cdots C_{T+1}^2 \alpha_k^2.$$

$$\tag{71}$$

The subsequent analysis builds on the following Lyapunov function:

$$\mathcal{V}_k := F(\theta_k) - F(\theta_*) + \sum_{l=1}^{T}\mathbb{E}\left[\left\|y_k^{(l)} - f^{(l)}(y_k^{(l-1)})\right\|^2\right] \tag{72}$$

where $\theta_*$ is the optimal solution of the problem.

Conditioning on $\mathcal{F}_k$ and taking expectation of $\mathcal{V}_{k+1}$ w.r.t. $\xi_k$, we have

$$
\begin{aligned}
\mathbb{E}\left[\mathcal{V}_{k+1}|\mathcal{F}_k\right] \leq & \mathcal{V}_k - \alpha_k\left(1 - \frac{\alpha_k}{2\beta_k}\sum_{l=1}^{T}A_l^2\right)\mathbb{E}\left[\|\nabla F\left(\theta_k\right)\|^2\right] + \frac{L}{2}C_1^2\cdots C_{T+1}^2\alpha_k^2 \\
& + (1+\beta_k)\sum_{l=1}^{T}\mathbb{E}\left[\left\|y_{k+1}^{(l)} - f^{(l)}\left(y_{k+1}^{(l-1)}\right)\right\|^2|\mathcal{F}_k\right] - \sum_{l=1}^{T}\mathbb{E}\left[\left\|y_k^{(l)} - f^{(l)}\left(y_k^{(l-1)}\right)\right\|^2|\mathcal{F}_k\right] \\
& - \frac{\beta_k}{2}\sum_{l=1}^{T}\mathbb{E}\left[\left\|y_{k+1}^{(l)} - f^{(l)}\left(y_{k+1}^{(l-1)}\right)\right\|^2|\mathcal{F}_k\right] \\
\overset{\text{(Lemma 8)}}{\leq}\ & \mathcal{V}_k - \alpha_k\left(1 - \frac{\alpha_k}{2\beta_k}\sum_{l=1}^{T}A_l^2\right)\mathbb{E}\left[\|\nabla F\left(\theta_k\right)\|^2\right] + \frac{L}{2}C_1^2\cdots C_{T+1}^2\alpha_k^2 \\
& + ((1+\beta_k)(1-\beta_k)-1)\sum_{l=1}^{T}\mathbb{E}\left[\left\|y_k^{(l)} - f^{(l)}(y_k^{(l-1)})\right\|^2|\mathcal{F}_k\right] \\
& + 4(1+\beta_k)(1-\beta_k)^2C_1^2\mathbb{E}\left[\|\theta_k - \theta_{k-1}\|^2|\mathcal{F}_k\right] \\
& + \sum_{l=2}^{T}\left[4(1+\beta_k)(1-\beta_k)^2C_l^2 + \gamma_l\right]\mathbb{E}\left[\left\|y_{k+1}^{(l-1)} - y_k^{(l-1)}\right\||\mathcal{F}_k\right] \\
& + 2(1+\beta_k)\beta_k^2NV^2 + \frac{1+\beta_k}{\beta_k}\prod_{j=k-t}^{k-1}(1-\beta_j)^2\sum_{l=1}^{T}\mathbb{E}\left[\|u_k^{(l)}\|^2\right] \\
& - \sum_{l=2}^{T}\gamma_l\mathbb{E}\left[\left\|y_{k+1}^{(l-1)} - y_k^{(l-1)}\right\||\mathcal{F}_k\right] - \frac{\beta_k}{2}\sum_{l=1}^{T}\mathbb{E}\left[\left\|y_{k+1}^{(l)} - f^{(l)}(y_{k+1}^{(l-1)})\right\|^2\right] \\
\leq & \mathcal{V}_k - \alpha_k\left(1 - \frac{\alpha_k}{2\beta_k}\sum_{l=1}^{T}A_l^2\right)\|\nabla F\left(\theta_k\right)\|^2 + \frac{L}{2}C_1^2\cdots C_{T+1}^2\alpha_k^2 + 2(1+\beta_k)\beta_k^2NV^2 \\
& + 4C_1^2\mathbb{E}\left[\|\theta_k - \theta_{k-1}\|^2|\mathcal{F}_k\right] \\
& + \sum_{l=2}^{T}(4C_l^2 + \gamma_l)\mathbb{E}\left[\left\|y_{k+1}^{(l-1)} - y_k^{(l-1)}\right\|^2|\mathcal{F}_k\right] \\
& - \sum_{l=2}^{T}\gamma_l\mathbb{E}\left[\left\|y_{k+1}^{(l-1)} - y_k^{(l-1)}\right\||\mathcal{F}_k\right] - \frac{\beta_k}{2}\sum_{l=1}^{T}\mathbb{E}\left[\left\|y_{k+1}^{(l)} - f^{(l)}(y_{k+1}^{(l-1)})\right\|^2\right] \\
& + \frac{1+\beta_k}{\beta_k}\prod_{j=k-t}^{k-1}(1-\beta_j)^2\sum_{l=1}^{T}\mathbb{E}\left[\|u_k^{(l)}\|^2\right]
\end{aligned}
$$

$$
\stackrel{\text{(Lemma 7)}}{\leq} \mathcal{V}_k - \alpha_k \left(1 - \frac{\alpha_k}{2\beta_k} \sum_{l=1}^{T} A_l^2\right) \mathbb{E}\left[\|\nabla F(\theta_k)\|^2\right] + \frac{L}{2} C_1^2 \cdots C_{T+1}^2 \alpha_k^2
$$

$$
+ 4C_1^2 \mathbb{E}\left[\|\theta_k - \theta_{k-1}\|^2 \,|\mathcal{F}_k\right]
$$

$$
\underbrace{+3\left(\frac{\beta_k}{1-\beta_k}\right)^2 \sum_{l=2}^{T} \left(4C_l^2 + \gamma_l\right) \mathbb{E}\left[\left\|y_{k+1}^{(l-1)} - f^{(l)}\left(y_{k+1}^{(l-2)}\right)\right\|^2\right]}_{:=I_{k_1}}
$$

$$
\underbrace{-\frac{\beta_k}{2} \sum_{l=1}^{T} \mathbb{E}\left[\left\|y_{k+1}^{(l)} - f^{(l)}\left(y_{k+1}^{(l-1)}\right)\right\|^2\right]}_{:=I_{k_2}}
$$

$$
\underbrace{+3\sum_{l=2}^{T} \left(4C_l^2 + \gamma_l\right) C_{l-1}^2 \mathbb{E}\left[\left\|y_{k+1}^{(l-2)} - y_k^{(l-2)}\right\|^2 \,|\mathcal{F}_k\right]}_{:=I_{k_3}}
$$

$$
\underbrace{-\sum_{l=2}^{T} \gamma_l \mathbb{E}\left[\left\|y_{k+1}^{(l-1)} - y_k^{(l-1)}\right\| \,|\mathcal{F}_k\right]}_{:=I_{k_4}}
$$

$$
+ \left(2\left(1 + \beta_k\right)\beta_k^2 N + \left(\frac{\beta_k}{1-\beta_k}\right)^2 \sum_{l=2}^{T} \left(4C_l^2 + \gamma_l\right)\right) V^2
$$

$$
+ \frac{1 + \beta_k}{\beta_k} \prod_{j=k-t}^{k-1} (1 - \beta_j)^2 \sum_{l=1}^{T} \mathbb{E}\left[\|u_k^{(l)}\|^2\right] + 3 \prod_{j=k-t}^{k-1} (1 - \beta_j)^2 \sum_{l=2}^{T} \left(4C_l^2 + \gamma_l\right) \mathbb{E}\left[\left\|u_k^{(l-1)}\right\|^2\right]
$$

$$
\tag{73}
$$

If we define $\gamma_l$ such that:

$$
3\left(\frac{\beta_k}{1-\beta_k}\right)^2 \sum_{l=2}^{T} \left(4C_l^2 + \gamma_l\right) < \frac{\beta_k}{2} \tag{74}
$$

and

$$
3\left(4C_l^2 + \gamma_l\right) C_{l-1}^2 < \gamma_{l-1}, \tag{75}
$$

then

$$
I_{k_1} + I_{k_2} < 0 \tag{76}
$$

$$
I_{k_3} + I_{k_4} < 0. \tag{77}
$$

It follows that

$$
\mathbb{E}\left[\mathcal{V}_{k+1}|\mathcal{F}_k\right] \leq \mathcal{V}_k - \alpha_k \left(1 - \frac{\alpha_k}{2\beta_k} \sum_{l=1}^{T} A_l^2\right) \mathbb{E}\left[\|\nabla F(\theta_k)\|^2\right] + \frac{L}{2} C_1^2 \cdots C_{T+1}^2 \alpha_k^2
$$

$$
+ \left(3(4C_2^2 + \gamma_2) + 4\right) C_1^2 \mathbb{E}\left[\|\theta_k - \theta_{k-1}\|^2 \,|\mathcal{F}_k\right]
$$

$$
+ \left(2\left(1 + \beta_k\right)\beta_k^2 N + \left(\frac{\beta_k}{1-\beta_k}\right)^2 \sum_{l=2}^{T} \left(4C_l^2 + \gamma_l\right)\right) V^2
$$

$$
+ \frac{1 + \beta_k}{\beta_k} \prod_{j=k-t}^{k-1} (1 - \beta_j)^2 \sum_{l=1}^{T} \mathbb{E}\left[\|u_k^{(l)}\|^2\right] + 3 \prod_{j=k-t}^{k-1} (1 - \beta_j)^2 \sum_{l=2}^{T} \left(4C_l^2 + \gamma_l\right) \mathbb{E}\left[\left\|u_k^{(l-1)}\right\|^2\right]
$$

$$
\tag{78}
$$

Setting $\beta_k = \alpha_k \sum_{l=1}^{T-1} A_l^2$, we have

$$
\begin{aligned}
\mathbb{E}\left[\mathcal{V}_{k+1}|\mathcal{F}_k\right] \leq & \mathcal{V}_k - \frac{\alpha_k}{2}\mathbb{E}[\|\nabla F\left(\theta_k\right)\|^2] + \frac{L}{2}C_1^2\cdots C_{T+1}^2\alpha_k^2 + (3(4C_2^2+\gamma_2)+4)C_1^2\mathbb{E}\left[\|\theta_k - \theta_{k-1}\|^2|\mathcal{F}_k\right] \\
& + \left(2\left(1+\beta_k\right)\beta_k^2 N + \left(\frac{\beta_k}{1-\beta_k}\right)^2\sum_{l=2}^{T}\left(4C_l^2+\gamma_l\right)\right)V^2 \\
& + \frac{1+\beta_k}{\beta_k}\prod_{j=k-t}^{k-1}(1-\beta_j)^2\sum_{l=1}^{T}\mathbb{E}\left[\left\|u_k^{(l)}\right\|^2\right] + 3\prod_{j=k-t}^{k-1}(1-\beta_j)^2\sum_{l=2}^{T}\left(4C_l^2+\gamma_l\right)\mathbb{E}\left[\left\|u_k^{(l-1)}\right\|^2\right]
\end{aligned}
\tag{79}
$$

Setting $\alpha_k = \alpha = \frac{c_\alpha}{\sqrt{K}}$ and $\beta_k = \beta = \frac{c_\beta}{\sqrt{K}}$ and summing up both sides of (79) from $k=0$ to $K-1$, we have

$$
\begin{aligned}
\frac{1}{K}\sum_{k=0}^{K-1}\|\nabla F\left(\theta_k\right)\|^2 \leq & \frac{2\mathcal{V}_0}{K\alpha} + LC_1^2\cdots C_{T+1}^2\alpha + 2(3(4C_2^2+\gamma_2)+4)C_1^4C_2^2\cdots C_{T+1}^2\alpha \\
& + \frac{2}{\alpha}\left(2\left(1+\beta\right)\beta^2 N + \left(\frac{\beta}{1-\beta}\right)^2\sum_{l=2}^{T}\left(4C_l^2+\gamma_l\right)\right)V^2 \\
& + \frac{2(1+\beta)\left(1-\beta\right)^{2(t-1)}}{\alpha\beta}\frac{1}{K}\sum_{k=0}^{K-1}\sum_{l=1}^{T}\mathbb{E}\left[\left\|u_k^{(l)}\right\|^2\right] \\
& + \frac{6\left(1-\beta\right)^{2(t-1)}}{\alpha}\frac{1}{K}\sum_{k=0}^{K-1}\sum_{l=2}^{T}\left(4C_l^2+\gamma_l\right)\mathbb{E}\left[\left\|u_k^{(l-1)}\right\|^2\right] \\
\overset{\text{(Assumption 5)}}{\leq} & O\left(\frac{1}{\sqrt{K}}\right) \\
& + \frac{2(1+\beta)\left(1-\beta\right)^{2(t-1)}}{\alpha\beta}\sum_{l=1}^{T}D_l^2 \\
& + \frac{6\left(1-\beta\right)^{2(t-1)}}{\alpha}C_u\sum_{l=1}^{T}D_l^2
\end{aligned}
\tag{80}
$$

where $C_u = \max\left(4C_l^2+\gamma_l\right)$ for $l = 1\ldots T$.

This completes the proof of Theorem 2. If we can set $t$ such that $\left(1-\beta\right)^{2(t-1)} = O(\beta^3)$, the algorithm achieves $O(1/\sqrt{K})$ convergence rate.

