# OpenReview forum: "SpSC: A Fast and Provable Algorithm for Sampling-Based GNN Training"
_ICLR.cc/2022/Conference — ICLR 2022 Submitted_

### Official Review · Reviewer_YqXv · 2021-10-31

**Correctness:** 3
**Technical Novelty And Significance:** 2
**Empirical Novelty And Significance:** 2
**Recommendation:** 3
**Confidence:** 4

**Main Review:**

This paper develops a variant of SCO algorithms using a sparse moving average. The resulting method can be applied to large-scale graph training with sampling-based methods. My main concerns of this paper are the novelty and the unconvincing experiments.

#####Pros#####

1. This work considers the GNN training problem of large-scale graphs. The studied problem is quite interesting and really useful for real-world applications.


2. The organization of this manuscript is good and easy to follow.

#####Cons#####

1. The main concern is that the novelty of this work is not enough. It is presented in previous works [Cong et al., 2020; 2021] that sampling-based GNN training can be understood as an SCO problem. Considering this point and the existence of SCGD work [Yang et al., 2019], this work only introduces the sparse version of the moving average, which can hardly be considered as a novel idea. In addition, the advantages in terms of time and memory of this method over SCO are not obvious, as shown in Table 3 and Figure 4. Therefore, the novelty and advantages of the proposal are limited.



2. The second concern is about the experiments. (1) This work only compares with original GCN and GraphSAGE, and does not include many recent methods on large-scale graph training, such as GNNAutoScale [1]. As far as I know, the performance of this method is not as good as the recent methods. (2) The current version does not provide a clear explanation of why Sparse_SCO can be better than SCO. The authors mention that it could be over-fitting. However, according to the training loss in Figure 3, we cannot observe obvious phenomenon of over-fitting. Hence, this point should be clarified.


[1] Fey, Matthias, et al. "GNNAutoScale: Scalable and Expressive Graph Neural Networks via Historical Embeddings." arXiv preprint arXiv:2106.05609 (2021).

**Summary Of The Paper:**

This paper studies the problem of training GNNs on large-scale graphs. Specifically, it proposes to use sparse moving average for sampling-based GNN training strategies. The convergence rate of the proposed approach has been proved under several assumptions. The authors conduct experiments on several large-scale datasets to show the effectiveness and efficiency of the proposal.

**Summary Of The Review:**

Overall, I think the current version is not ready for publishing on ICLR due to the limited novelty and unconvincing experiments. I recommend authors to strengthen its novelty and make the experiments to be more convincing.

---

> ### Author Response · Authors · 2021-11-22
> **response**
>
> Q1: The advantages in terms of time and memory of this method over SCO are not obvious.
> A1: As shown in Figure4, the advantage of our algorithm in terms of memory consumption is obvious on large graphs. Besides time and memory, compared to SCO, our proposed method shows higher accuracy on every graph (Table2).
>
> Q2: This work does not include many recent methods on large-scale graph training, such as GNNAutoScale.
> A2: Thank you for the reference. We will add comparisons with these recent methods.

---

### Official Review · Reviewer_7iri · 2021-10-31

**Correctness:** 2
**Technical Novelty And Significance:** 2
**Empirical Novelty And Significance:** 2
**Recommendation:** 3
**Confidence:** 5

**Main Review:**

Pros:
1. The idea of reducing the memory cost is interesting. It does help to apply the stochastic compositional gradient descent method to train GNN.

2. The experimental results show improvement over existing methods.

Cons:

1. The writing is not clear, e.g. what's the meaning of the last term in Eq.(11)? More explanation is needed.

2. The definition for the projection matrix is not very clear. Is it a  matrix with binary values?

3. The proof is not correct. If I am right, you cannot guarantee $\lambda_1>0$ in Theorem 1. Hence, the convergence rate is not true.

4. Assumption 4 is not reasonable. No literature about the stochastic compositional optimization problem used this assumption.  According to the value of $\beta_k$ in Theorem 1, when $K$ approaches infinity, the variance approaches zero. It is not reasonable. Even though the full gradient approaches zero when it converges, we cannot say the stochastic gradient approaches zero. Thus, the variance could be large. I don't think this assumption is reasonable.

5. Given Assumption 4, why is Assumption 6 introduced?

=======after response=====

Thanks for your response. But there are still some fatal flaws in the theoretical analysis.

1. Assumption 4 is still NOT reasonable. Considering that we just use a small subset of neighboring nodes, then the variance is large. But, based on Assumption 4, the variance could approach zero when $\beta$ approaches zero. Hence, this assumption is definitely NOT reasonable.

2. Theorem 1 still fails to provide a feasible $\beta$. The authors give an inequality constraint: $\beta\leq f(\beta)$. But it is unclear whether this inequality has a feasible solution.

**Summary Of The Paper:**

This paper formulated the graph neural network as a stochastic compositional optimization problem and then developed a new optimization algorithm to train GNN. However, there are some errors. It is not ready for publication.

**Summary Of The Review:**

The idea of reducing the memory cost is interesting and useful. However, there are so many flaws in the theoretical analysis. Hence, I recommend rejection.

---

> ### Author Response · Authors · 2021-11-22
> **response**
>
> Q1: What’s the meaning of the last term in Eq.(11).
> A1: Since we only store the moving averages of nodes that are sampled in the most recent t iterations, the information of the overwritten nodes is lost. The last term in Eq.(11) defines the information of the overwritten nodes.  We have added more descriptions in the updated version.
>
> Q2: The definition for the projection matrix is not very clear. Is it a matrix with binary values?
> A2: Yes, this is a matrix with binary values. It indicates the nodes that are overwritten in the current iteration, i.e. the nodes that are sampled in iteration k−t and are not sampled in the following t iterations.
>
> Q3: The convergence rate is not true if you cannot guarantee \lambda_1>0 in Theorem 1.
> A3: We can always configure $\alpha$ and $\beta$ to ensure $\lambda_1>0$. We skipped this step in our initial submission. We have fixed it in the new version.
>
> Q4: Assumption 4 is not reasonable. We cannot say the stochastic gradient approaches zero.
> A4: Thank you for pointing it out. There is a typo in this assumption. There should not be $\nabla$s. We have fixed it in the new version. The assumption says the estimation variance of the composite function is small enough. It is a reasonable assumption because when the number of neighbors is large enough, the estimation variance does approach zero. The assumption is used in previous SCO analysis (e.g. Balasubramanian et al., 2020).
>
> Q5: Given Assumption 4, why is Assumption 6 introduced?
> A5: Assumption4 and Assumption6 are used for the proof of Theorem1 and Theorem2 separately.

---

### Official Review · Reviewer_xK4K · 2021-11-02

**Correctness:** 3
**Technical Novelty And Significance:** 2
**Empirical Novelty And Significance:** 2
**Recommendation:** 5
**Confidence:** 3

**Main Review:**

Strengths:
1. The SCO perspective of sampling-based GNN training seems an interesting direction. The idea of improving existing SCO algorithms by only storing the data for nodes sampled in the past few iterations in SCO algorithms is simple and effective.
2. Compared to SCO and Adam-sample, SpSC has overall better performance.
3. Under certain assumptions, SpSC has non-trivial convergence guarantees, which is an advantage against many sampling-based approaches.

Weaknesses:
1. I think the technical novelty of SpSC is not significant. The difference between SCGD and SpSC is incremental. Also the analysis on convergence mainly follows the framework introduced in prior work.
2. I find that the description of SpSC is difficult to understand. The convergence analyses are quite complicated, and I think some intuition should be provided.
3. The convergence results depends on many assumptions. I think more comments on these assumptions should be helpful.
4. In the experiments, it seems that the time and space consumptions of SCO are comparable to those of SpSC, except for the product dataset. Can you provides an explanation on this phenomenon?
5. There has been lots of recent work on scalable training of GNNs. I think the authors should compare more baselines in the experiments, e.g., VR-GCN, GraphSAINT, or more recently AutoScale GNN. Otherwise, the value and impact of SpSC is unclear.

**Summary Of The Paper:**

This paper studies neighbor sampling techniques for training GNNs. Previous work has observed that sampling-based GNN training can be formulated as a Stochastic Compositional Optimization (SCO) problem. The authors argue that naive implementation of existing SCO algorithms incurs huge memory cost for training GNNs. This paper propose a Sparse Stochastic Compositional (SpSC) gradient method, which only stores the data for nodes sampled in the past few iterations. Convergence analysis on SpSC is provided and empirical results show that SpSC has better performance than naive SCO.

**Summary Of The Review:**

Overall, I think the SCO perspective is interesting, and SpSC demonstrates the possibility of using SCO algorithms to improve the scalability of sampling-based training methods. However, the technical novelty may not be enough. Moreover, I think authors should compare more baselines in the experiments and the presentation could also be improved.

---

> ### Author Response · Authors · 2021-11-22
> **response**
>
> Q1: The description of SpSC is difficult to understand. Some intuition should be provided.
> A1: We have simplified the proof and added more descriptions to our algorithm and proof in the updated version.
>
> Q2: The convergence results depend on many assumptions, and need more comments.
> A2: Assumption1-4 are the standard assumptions for SCO analysis (e.g. Yang et al., 2019; Balasubramanian et al., 2020). We only added assumption5, which is reasonable in the context of GNN training.
>
> Q3: Why the space consumptions of SCO are comparable to those of SpSC?
> A3: We made a mistake and measured the memory consumption including the validation procedure. We have recollected the memory consumption from the training process only and updated the results in Figure4. The results show that, on small graphs, such as arxiv, reddit, and proteins, the memory consumption of SCO is a little higher than SpSC. However, on larger graphs, such as products and yelp, SpSC uses much less memory than SCO.

---

### Official Review · Reviewer_pkB7 · 2021-11-08

**Correctness:** 4
**Technical Novelty And Significance:** 2
**Empirical Novelty And Significance:** 2
**Recommendation:** 3
**Confidence:** 3

**Main Review:**

### Strengths:
1. This paper has done a thorough theoretical analysis of convergence. And the experimental results are presented in detail.

### Questions:
1. Why is the performance of Adam_Sample with GCN on ogbn-arxiv (Table 2) much lower than the Adam_Full's? Since in the last paragraph of Section 5 and in Figure 4, it is explained that `For arxiv, since it is a small graph, full neighbor aggregation has almost the same memory consumption as sampled neighbor aggregation.` If the sampling rate is large for the arxiv dataset and almost all neighbors are considered, the performance should be close to Adam_Full.
2. It seems that the hit rate (the ratio of sample nodes that can be found in the buffer) can be significantly affected by the sampling strategy. For example, the hit rate might be low if randomly sampling nodes on a large graph, while it might be relatively high if the center nodes are sampled through a random walk (there are lots of common neighbors of two neighboring nodes). Do we need to control the hit rate for the theoretical results (it seems that none of the assumptions is about this)? And how does the hit rate affect the efficiency/performance? In Figure 3, it is not evident that using larger $t$ (thus larger hit rate) can bring any gain in performance/convergence speed.

### Weaknesses:
1. The paper did not compare the proposed algorithm with the state-of-the-art sampling strategies. In section 5, it is said that `We adopt the layer-wise sampling method in Zou et al. (2019) for neighbor sampling.` I assume this layer-wise sampling strategy is used for the Adam_Sample baseline and the proposed Sparse_SCO algorithm (please correct me if wrong). However, it is known that some other subgraph sampling strategies may be better on the considered datasets in terms of performance (e.g., on YELP, GraphSAINT-RW can achieve 0.653±0.003; see Table 2 in the GraphSAINT paper (Zeng et al., 2019), while the Adam_Sample reported is 0.631) and convergence speed & training time (e.g., Figure 2 in (Zeng et al., 2019)). In general, given that there are many sampling strategies available now, considering only one sampling method (as baselines and used in Sparse_SCO) is insufficient. The paper said, `it is hard to draw a direct comparison with GraphSAINT because the sampling methods and the model architectures are different.` However, GraphSAINT can be applied to any model architecture. And it is not only when the sampling method is exactly the same, the two scalable approaches are comparable.
2. The paper considered limited GNN backbone architectures/aggregators. Firstly, because of Assumption 3, i.e., sampled neighbor aggregation is unbiased, the aggregator considered might be limited to some elementary ones, e.g., mean aggregator in GCN and GraphSAGE. However, even if it is hard to be incorporated into the theoretical framework, it is interesting to try other aggregators in GIN, GAT, and PNA. The practical usefulness might be limited if the algorithm cannot be applied to these famous GNN backbones.

**Summary Of The Paper:**

This paper proposed an improved variant of the Stochastic Compositional Optimization (SCO) framework to train GNNs, replacing all nodes' moving averages with a sparse representation. The proposed algorithm only requires a fixed-size buffer, regardless of the graph size, solving the memory issue of SCO algorithms and making it practically applicable to large graphs. The paper showed that the proposed algorithm preserves the convergence rate of the original SCO algorithm and experimentally validated that the algorithm could outperform the traditional Adam SGD for GNN training with a small memory overhead.

**Summary Of The Review:**

This paper proposed an improved variant of the Stochastic Compositional Optimization (SCO) framework to train GNNs, using a memory buffer to approximate the moving averages of all nodes. Such a fixed-size buffer solution makes the SCO algorithms practical for large graphs. However, limited intuitions are provided to understand the design of such buffer and its effectiveness in practice. Most importantly, the proposed Sparse_SCO algorithm is not superior to Adam_Sample in terms of memory and time efficiency. The performance and convergence speed are better than layer-wise sampling method (Zou et al., 2019), but it is questionable whether it can outperform other sampling strategies like GraphSAINT (Zeng et al., 2019) and ClusterGCN (Chiang et al., 2019). Moreover, the proposed SpSC algorithm is only evaluated with GCN and GraphSAGE and using one sampling algorithm in (Zou et al., 2019). It is unclear if it can be applied to other GNN backbone models and combined with other sampling strategies. Given the reasons above, I cannot recommend the current manuscript for acceptance.

---

> ### Author Response · Authors · 2021-11-22
> **response**
>
> Q1: Why Adam_Sample converges much slower than Adam_Full but the memory consumption is similar?
> A1: We made a mistake when measuring memory consumption. We included the validation procedure by mistake. Since we used large batch for validation, the total memory consumption of Adam_Sample and Adam_Full are similar. We recollected the memory consumption for the training process only and updated the results in Figure4. The results show that memory consumption of Adam_Full is much higher than the Adam_Sample.
>
>
> Q2: Do we need to control the hit rate for the theoretical results? Lack assumptions about hit rate.
> A2: No, we do not make any assumption about hit rate. The convergency result is not related to the hit rate.

---

> > ### Comment · Reviewer_pkB7 · 2021-12-01
> > **Thanks for the response**
> >
> > Thanks for your response. I am sorry for my late reply. I have carefully read the response and the other reviewer's comments. The authors have answered two of my questions but have not addressed the weak points I listed. Thus the reasons for not recommending the current manuscript are not changed, and I would like to maintain my score.

---

### Author Response · Authors · 2021-11-22
**response**

We thank the reviewers for the helpful comments and suggestions. We realize that there are some errors in our writing and experimental configurations. We have fixed the issues and highlighted the changes in the updated version. We will take the suggestions and add comparisons with more recent training methods on more GNN models to improve our experiments.

---

### Decision · Program_Chairs · 2022-01-20

**Decision:**

Reject

**Comment:**

This work proposes to train large-scale graph neural networks by replacing the moving averages used in the stochastic compositional optimization (SCO) framework with sparse moving averages. This reduces the memory required for SCO, allowing their algorithm to scale  to larger graphs.

The consensus is that the approach is reasonable, but incremental both in the change over SCGD and the change in the analysis. More importantly, the reviewers identified several sampling-based methods for scaling up training of GNNs that are important baselines for the proposed algorithm; the relative merits of the method against these approaches should be established with further experiments.